

# Anomalies in the space of coupling constants and their dynamical applications I

**Clay Córdova[1*], Daniel S. Freed[2], Ho Tat Lam[3] and Nathan Seiberg[1]**

**1** School of Natural Sciences, Institute for Advanced Study, Princeton NJ, USA
**2** Math Department, University of Texas at Austin, Austin, TX, USA
**3** Physics Department, Princeton University, Princeton, NJ, USA

* claycordova@ias.edu

## Abstract

It is customary to couple a quantum system to external classical fields. One application is to couple the global symmetries of the system (including the Poincaré symmetry) to background gauge fields (and a metric for the Poincaré symmetry). Failure of gauge invariance of the partition function under gauge transformations of these fields reflects 't Hooft anomalies. It is also common to view the ordinary (scalar) coupling constants as background fields, i.e. to study the theory when they are spacetime dependent. We will show that the notion of 't Hooft anomalies can be extended naturally to include these scalar background fields. Just as ordinary 't Hooft anomalies allow us to deduce dynamical consequences about the phases of the theory and its defects, the same is true for these generalized 't Hooft anomalies. Specifically, since the coupling constants vary, we can learn that certain phase transitions must be present. We will demonstrate these anomalies and their applications in simple pedagogical examples in one dimension (quantum mechanics) and in some two, three, and four-dimensional quantum field theories. An anomaly is an example of an invertible field theory, which can be described as an object in (generalized) differential cohomology. We give an introduction to this perspective. Also, we use Quillen's superconnections to derive the anomaly for a free spinor field with variable mass. In a companion paper we will study four-dimensional gauge theories showing how our view unifies and extends many recently obtained results.

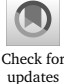

# 1  Introduction and Summary

't Hooft anomalies lead to powerful constraints on the dynamics and phases of quantum field theory (QFT). They also control the properties of boundaries, extended excitations like strings and domain walls, and various defects.

't Hooft anomalies do not signal an inconsistency of the theory. Instead, they show that some contact terms cannot satisfy the Ward identities of global symmetries. More generally, they are an obstruction to coupling the system to classical background gauge fields for these symmetries.

In this paper we generalize the notion of 't Hooft anomalies to the space of coupling constants. In addition to coupling the system to classical background gauge fields, we also make the various coupling constants spacetime dependent, i.e. we view them as background fields. The generalized 't Hooft anomalies are an obstruction to making the coupling constants and the various gauge fields spacetime dependent.

As with the ordinary 't Hooft anomalies, we use these generalized anomalies to constrain the phase diagram of the theory as a function of its parameters and to learn about defects constructed by position-dependent coupling constants.

## 1.1 Anomalies and Symmetries

A useful point of view of 't Hooft anomalies is to couple a system with a global symmetry to an appropriate background gauge field $A$. Here $A$ denotes a fixed classical source and leads to a partition function $Z[A]$. Depending on the context, $A$ could be a standard background connection for an ordinary continuous (0-form) global symmetry, or an appropriate background field for more subtle concepts of symmetry such as a discrete gauge field for a discrete global symmetry, a higher-form gauge field for a higher-form symmetry [1], or a Riemannian metric for (Wick rotated) Poincaré symmetry. Additionally, the partition function may depend on discrete topological data such as a choice of spin structure in a theory with fermions or an orientation on spacetime. We will denote all this data by $A$.

Naively one expects that the resulting partition function $Z[A]$ should be gauge invariant under appropriate background gauge transformations. An 't Hooft anomaly is a mild violation of this expectation. Denoting a general gauge transformation with gauge parameter $\lambda$ (or coordinate transformation) as $A \to A^\lambda$, the partition function $Z[A]$ is in general not gauge invariant. Instead, it transforms by a phase, which is a local functional of the gauge parameter $\lambda$ and the gauge fields $A$

$$Z[A^\lambda] = Z[A] \exp\left( -2\pi i \int_X \alpha(\lambda, A) \right) , \tag{1}$$

where $X$ is our $d$-dimensional spacetime.

The partition function $Z[A]$ is subject to a well-known ambiguity. Different regularization schemes can lead to different answers. This ambiguity can be absorbed in adding local counterterms to the action. These counterterms can depend on the dynamical fields and on background sources. This freedom in adding counterterms is the same as performing a redefinition in the space of coupling constants. A special case of such counterterms are those that multiply the unit operator, i.e. they depend only on classical backgrounds $A$. We refer to these terms as classical counterterms or sometimes simply as counterterms when the context is clear. An essential part of our discussion will involve such classical counterterms. The 't Hooft anomaly for the global symmetry is what remains of the phase in (1) after taking into account this freedom.[1]

---

[1]As noted in [2], one can always remove the anomalous phase by adding a $d$-form background field $A^{(d)}$ with a coupling $i\int_X A^{(d)}$. $A^{(d)}$ can be thought of as a background gauge field for a "$d-1$-form symmetry" that does not act on any dynamical field. (Such couplings are common in the study of branes in string theory.) Then the anomaly is removed by postulating that under gauge transformations of the background fields it transforms as $A^{(d)} \to A^{(d)} + d\lambda^{(d-1)} - 2\pi i \alpha(\lambda, A)$. The term with $\lambda^{(d-1)}$ is the standard gauge transformation of such a gauge field and the term with $\alpha$, which cancels (1), reflects a higher-group symmetry. See e.g. [2–4] and references therein.

Thus, the set of possible 't Hooft anomalies for a given global symmetry is defined by a cohomology problem of local phases consistent with the equation (1) modulo the variation of local functionals of the gauge field $A$.

It is convenient to describe anomalies using a classical, local action for the gauge fields $A$ in $(d+1)$-spacetime dimensions. Such actions are also referred to as invertible field theories.[2] In this presentation the $d$-dimensional manifold $X$ supporting the dynamical field theory is viewed as the boundary of a $(d+1)$-manifold $Y$, and we extend the classical gauge field sources $A$ to the manifold $Y$. On $Y$ there is a local, classical Lagrangian $-2\pi i\omega(A)$ with the property that

$$\exp\left(2\pi i\int_Y \omega(A^\lambda) - 2\pi i\int_Y \omega(A)\right) = \exp\left(2\pi i\int_X \alpha(\lambda,A)\right) . \tag{2}$$

Thus on closed $(d+1)$-manifolds the action $\omega$ defines a gauge-invariant quantity, while on manifolds with boundary it reproduces the anomaly.[3] We refer to $\omega(A)$ as the Lagrangian of the anomaly theory and we define the partition function of the anomaly theory as

$$\mathcal{A}[A] = \exp\left(2\pi i\int \omega(A)\right) . \tag{3}$$

Using these observations, we can present another point of view on the partition function of a theory with an 't Hooft anomaly. We can introduce a modified partition function as follows:

$$\tilde{Z}[A] \equiv Z[A]\exp\left(2\pi i\int_Y \omega(A)\right) . \tag{4}$$

In (4), the manifold $Y$ is again an extension of spacetime. Using the transformation law (1) and the definition (2) of $\omega$ we conclude that the partition function is exactly gauge invariant

$$\tilde{Z}[A^\lambda] = \tilde{Z}[A] . \tag{5}$$

The price we have paid is that the partition function now depends on the extension of the classical fields into the bulk. In some condensed matter applications, this added bulk $Y$ is physical. The system $X$ is on a boundary of a space $Y$ in a non-trivial SPT phase. The 't Hooft anomaly of the boundary theory is provided by inflow from the nontrivial bulk $Y$. This is known as anomaly inflow and was first described in [5]. (See also [6].)

Although the partition functions $Z$ and $\tilde{Z}$ are different, an essential observation is that, for $A = 0$, they encode the same correlation functions at separated points. It is this data that we view as the intrinsic defining information of a quantum field theory. However, one advantage of the presentation of the theory using $\tilde{Z}$ is that it clarifies the behavior of the anomaly under renormalization group flow.

First, such a transformation can modify the scheme used to define the theory in a continuous fashion. This means that in general, $d$-dimensional counterterms are modified along the flow. Second, we can also ask about the behavior of the classical anomaly action $\omega$ which resides in $(d+1)$ dimensions. If we view this term as arising from the long distance behavior of massive degrees of freedom (a choice of scheme) then along renormalization group

---

[2]In condensed matter physics, symmetry protected topological orders (SPTs) are also characterized at low energies by such actions. Depending on the precise definitions and context, "SPT" may be synonymous with "invertible field theory", or may instead refer to the deformation class of an invertible field theory, i.e. the equivalence class of invertible theories obtained by continuously varying parameters.

[3]In certain cases, there is no $Y$ such that $\partial Y = X$ and $A$ on $X$ is extended into $Y$. Then, one can construct an anomaly free partition function by assuming that $X$ is a component of the boundary of $Y$ and $Y$ has additional boundary components.

flow we can continuously adjust the details of these heavy degrees of freedom and hence $\omega$ could evolve continuously as well. Thus, a renormalization group invariant quantity is the deformation class of the action $\omega$, i.e. all actions that may be obtained from $\omega$ by continuous deformations.

In the applications to follow, we therefore focus on physical conclusions that depend only on the deformation class of $\omega$. (We will also see that theories related by renormalization group flow can produce different expressions for $\omega$ in the same deformation class). In particular, any theory with an anomaly action $\omega$ that is not continuously connected to the trivial action cannot flow at long distances to a trivially gapped theory with a unique vacuum and no long-range degrees of freedom.[4] It is this feature of 't Hooft anomalies that makes them powerful tools to study the dynamics of quantum field theories. We will revisit these general ideas in section 1.6 below.

In this paper, we generalize the notions above to the space of parameters of a QFT. We will describe how certain subtle phenomena can be viewed as a generalization of the concept of anomalies from the arena of global symmetries to this broader class of sources. In particular we will see how such anomalies of $d$-dimensional theories can also be summarized in terms of classical theories in $d + 1$-dimensions. We will use this understanding to explore phase transitions as the parameters vary and properties of defects that are associated with spacetime dependent coupling constants.

Our analysis extends previous work on this subject in [7–11]. (For a related discussion in another context see e.g. [12].) Finally, we would like to point out that an anomaly in making certain coupling constant background superfields was discussed in [13, 14]. It would be nice to phrase these anomalies and ours in a uniform framework.

## 1.2 Anomalies in Parameter Space: Defects

Instead of phrasing the analysis above in terms of background fields, it is often convenient to formulate the discussion in terms of defects and extended operators. Indeed, an ordinary global symmetry implies the existence of codimension one operators that implement the symmetry action. This paradigm also extends to other forms of internal symmetry: for instance $p$-form global symmetries are encoded in extended operators of codimension $p + 1$ [1]. Geometrically, these symmetry defects are Poincaré dual to the flat background gauge fields described above. These extended operators have the property that they are topological: small deformations of their positions do not modify correlation functions.

Many of the implications of 't Hooft anomalies are visible when we consider correlation functions of these extended operators. In this context 't Hooft anomalies arise as mild violations (by phases) of the topological nature of the symmetry defects. This perspective points the way to a natural generalization of the concept of anomalies to the space of coupling constants in quantum field theories. We promote the parameters of a theory to be spacetime-dependent and explore the properties of the resulting topologically non-trivial extended objects.

An important example, which will occur repeatedly below, is a circle-valued parameter such as a $\theta$-angle in gauge theory. This can be made to depend on a single spacetime coordinate $x$, and wind around the circle as $x$ varies from $-\infty$ to $+\infty$ (or around a nontrivial compact cycle in spacetime). If the bulk theory is trivially gapped, i.e. does not even have topological order (as in e.g. 4d $SU(N)$ Yang-Mills theory), this leads to an effective theory in $d - 1$ spacetime dimensions. Depending on the profile of the parameter variation there are several possibilities for the physics (illustrated in Figure 1).

---

[4]A trivially gapped theory by definition has a gap in its spectrum of excitations and its long distance behavior is particularly simple. In particular, it has a single ground state on any space of finite volume. This means that it does not have even topological degrees of freedom at low energies. In this case the low-energy theory is a classical theory of the background fields also referred to as an invertible theory.

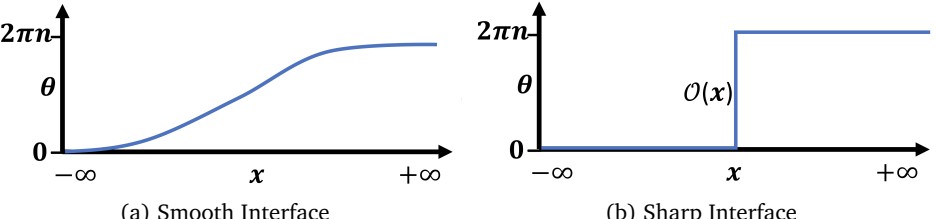

Figure 1: Interfaces defined by spatially varying coupling constant $\theta(x)$. In (a), the variation is smooth and the resulting interface dynamics are universal. In (b), the variation is abrupt. The resulting worldvolume dynamics is not universal and can be modified by coupling to degrees of freedom on the interface (schematically denoted $\mathcal{O}(x)$ above. As we will discuss, for certain special choices of $\mathcal{O}(x)$, an abrupt interface can be made completely transparent.

- If the parameter variation is smooth, i.e. it takes place over a distance scale longer than the UV cutoff, then the resulting interface dynamics is completely determined by the UV theory. It is universal.[5] In other words, these smooth interfaces are well defined observables of the QFT. Such interfaces have been widely studied for instance recently in $4d$ QCD and related applications to $3d$ dualities [15–18]. One of the main applications of our formalism is to give a systematic point of view on the worldvolume anomaly of such interfaces. In particular we will see how they may be obtained from inflow from the $d+1$-dimensional classical theory encoding the bulk anomaly in the space of parameters. We will refer to such interfaces as "smooth interfaces" or simply interfaces.

- If the parameter variation is abrupt, more precisely, if it takes place over a distance scale comparable to the UV cutoff, the dynamics on the interface depends on additional UV data. It is not universal. For instance, such a sharp interface can always be decorated by coupling it to a $(d-1)$-dimensional QFT. To illustrate the difference more explicitly, consider for instance including in the UV Lagrangian a term of the form

$$\delta\mathcal{L} = \frac{1}{\Lambda^{\Delta+1-d}}\partial^{\mu}\theta(x)V_{\mu}(x) , \tag{6}$$

where $\Lambda$ is a UV cutoff scale, and $V_{\mu}$ is an operator of scaling dimension $\Delta$. If $\Delta$ is sufficiently large, and the gradients $\partial^{\mu}\theta(x)$ are small compared to the cutoff scale then this term is irrelevant at large distances. (Dangerously irrelevant operators should be treated separately.) However, when the gradients $\partial^{\mu}\theta(x)$ are large terms such as (6) become relevant and the interface dynamics depends on their coefficients. We will refer to such interfaces as "sharp interfaces" or as defects.

- A special case of such a sharp interface is the following. If the parameter variation is completely localized and the discontinuity and $(d-1)$-dimensional theory are chosen appropriately, then the resulting interface can be made to be completely transparent. We will refer to these as "transparent interfaces." Such transparent interfaces will play a key role below.

We should emphasize that these interfaces should be distinguished from domain walls. Domain walls are also co-dimension one objects. But unlike the interfaces, they are dynamical

---

[5]Following standard terminology, universal properties of quantum field theories are independent of the details of the UV theory at the cutoff scale.

excitations. They interpolate between two degenerate ground states and can move around. In contrast, interfaces are pinned by the external variation of the parameters.

While real-valued parameters often lead to codimension one defects, complex-valued parameters are naturally associated with defects of codimension two. A characteristic example is a $4d$ Weyl fermion with a position dependent complex mass $m(x, y)$ depending on two spacetime coordinates and winding $n$ times around infinity. This example is the essence of the phenomenon investigated in [5,19,20]. The winding mass leads to two-dimensional Majorana-Weyl fermion zero modes localized at the zeros of the mass.

Below we will explain how this example can be viewed as an instance of anomalies in the space of masses. In particular, this means that the index theorem counting zero modes can be obtained by integrating an appropriate anomaly six-form (related to the $5d$ classical anomaly theory by descent):[6]

$$\mathcal{I}_6 = \frac{1}{384\pi^2}\gamma(m) \wedge \text{Tr}(R \wedge R) = \frac{1}{48}\gamma(m) \wedge p_1 \,, \tag{8}$$

where $\gamma(m)$ is a two-form on the mass parameter space with total integral one, and $p_1$ is the first Pontrjagin class of the manifold.[7] For instance, it is often natural to take $\gamma(m) = \delta^{(2)}(m)d^2m$.

One virtue of this presentation of the anomaly in the space of mass parameters is that they are manifestly robust under a large class of continuous deformations. For instance, we can deform the $4d$ free fermion by coupling it to any interacting theory preserving the large $|m|$ asymptotics. The anomaly (8) remains non-trivial and implies that in any such theory, $2d$ defects arising from position-dependent mass with winding number $n$ have chiral central charge, $c_L - c_R = n/2$.

## 1.3 Anomalies in Parameter Space: Families of QFTs

Another significant application of our techniques is to constraining the properties of families of QFTs. A typical situation we will consider is a family of theories labelled by a parameter such that at generic values of the parameter the theory is trivially gapped. An anomaly in the space of coupling constants can then imply that somewhere in the parameter space the infrared must be non-trivial.

An illustrative example that we describe in section 4 is two-dimensional $U(1)$ gauge theory coupled to $N$ scalar fields of unit charge. At long distances the theory is believed to generically have a unique ground state. However, this conclusion cannot persist for all values of the coupling constants: for at least one value $\theta_* \in [0, 2\pi)$ the infrared must be non-trivial, and hence there is a phase transition as $\theta$ is dialed through this point.

We will argue for this conclusion by carefully considering the periodicity of $\theta$. Placing the theory on $\mathbb{R}^2$ in a topologically trivial configuration of background fields, i.e. all those necessary to consider all correlation functions of local operators in flat space, the parameter

---

[6]In general spacetime dimension $d$ we can define an anomaly $(d + 2)$-form $\mathcal{I}_{d+2}$ by the property that the anomalous variation of the action ($\alpha(\lambda, A)$ above) is computed by

$$d\alpha(\lambda, A) = \delta\omega \,, \qquad d\omega = \mathcal{I}_{d+2} \,, \tag{7}$$

and above $\delta$ denotes the gauge variation. However, it is also possible for the anomaly $\omega(A)$ to be non-trivial and yet nevertheless the $(d + 2)$-form $\mathcal{I}_{d+2}$ vanishes.

[7]Compactifying the complex mass plane by adding the point at infinity, the parameter space is topologically a two-sphere. This means that the free $4d$ Weyl fermion gives an elementary example of a field theory with an effective two-cycle in the space of parameters. More complicated examples of such two-cycles in the parameter space involving M5 branes or electric-magnetic duality were recently discussed in [9, 10, 21] in connection with some earlier assertions in [22].

$\theta$ has periodicity $2\pi$. However when we couple to topologically non-trivial background fields the $2\pi$-periodicity is violated.

Specifically, this gauge theory has a $PSU(N) \cong SU(N)/\mathbb{Z}_N$ global symmetry. In the presence of general background fields $A$ for this global symmetry group the instanton number of the dynamical $U(1)$ gauge group can fractionalize. This means that in such configurations the periodicity of $\theta$ is enlarged to $2\pi N$. This violation of the expected periodicity of $\theta$ in the presence of background fields is conceptually very similar to the general paradigm of anomalies described in section 1.1. As in the discussion there, we find that the $2\pi$ periodicity of $\theta$ can be restored by coupling the theory to a three-dimensional classical field theory that depends on $\theta$. Its Lagrangian is[8]

$$\omega = \frac{1}{N} \frac{d\theta}{2\pi} \cup w_2(A) \,, \tag{9}$$

where $w_2(A) \in H^2(X, \mathbb{Z}_N)$ is the second Stiefel-Whitney class[9] measuring the obstruction of lifting an $PSU(N)$ bundle to an $SU(N)$ bundle. In particular, this non-trivial anomaly must be matched under renormalization group flow now applied to the family of theories labelled by $\theta \in [0, 2\pi)$. A trivially gapped theory for all $\theta$ does not match the anomaly and hence it is excluded.

We can also describe the anomaly and its consequences somewhat more physically as follows. The theories at $\theta = 0$ and $\theta = 2\pi$ differ in their coupling to background $A$ fields by a classical function of $A$ (a counterterm) $\frac{2\pi}{N} w_2(A)$ [23]. However, since the coefficient of this counterterm must be quantized, this difference cannot be removed by making its coefficient $\theta$-dependent in a smooth fashion. This means that at some $\theta_* \in [0, 2\pi)$ the vacuum must become non-trivial so that the counterterm can jump discontinuously. For instance in the special case $N = 2$ with a potential leading to a $\mathbb{CP}^1$ sigma model at low energies, the theory at $\theta = \pi$ is believed to be a gapless WZW model. For larger $N$, the $\mathbb{CP}^{N-1}$ model is believed to have a first order transition at $\theta = \pi$ with two degenerate vacua associated to spontaneously broken charge conjugation symmetry.

This example is emblematic of our general analysis below. We discuss QFTs with two essential properties. First, parameters can change continuously between two points with the same local physics. (In this example we shift the $\theta$-parameter by $2\pi$.) Second, the counterterms of background fields after the change are different. (In this example, the coefficient of the counterterm $w_2(A)$ is different.) Furthermore, if the coefficient of the counterterm is quantized, this difference cannot be eliminated by making its coefficient parameter-dependent in a continuous fashion. We interpret this as an 't Hooft anomaly in the space of parameters and other background fields.

Then, the low-energy theory must saturate that anomaly. If it is nontrivial, i.e. gapless or gapped and topological, it should have the same anomaly. And if it is gapped and trivial for generic values of the parameters, there must be a phase transition for some value of the parameters. The fact that discontinuities in counterterms require phase transitions is widely known and applied, here we see how to phrase this idea in terms of 't Hooft anomalies.[10]

The example of $U(1)$ gauge theories described above is also a good one to illustrate the relation of our discussion to previous analyses of anomalies of discrete symmetries such as time-reversal, T, and charge conjugation, $\mathcal{C}$, in these models discussed in [23, 36, 37]. These

---

[8]A non-expert physicist can think of the cup product as a version of a wedge product of differential forms appropriate for cohomology classes valued in finite groups.

[9]In the mathematics literature the Stiefel-Whitney classes are defined for principal O($N$)-bundles. The characteristic class which measures the obstruction to lifting a projective bundle to a vector bundle could be called a "Brauer class".

[10]For instance in the study of 3$d$ dualities the total discontinuity in various Chern-Simons levels for background fields as a parameter is varied is independent of the duality frame and provides a useful consistency check on various conjectures [24–35].

theories are $\mathsf{T}$ (and $\mathcal{C}$) invariant at the two values $\theta = 0$ and $\theta = \pi$. For even $N$ when $\theta = \pi$, there is a mixed anomalies between $\mathsf{T}$ (or $\mathcal{C}$) and the $PSU(N)$ global symmetry, and hence the long-distance behavior at $\theta = \pi$ cannot be trivial in agreement with the general discussion above. For odd $N$ the situation is more subtle. In this case there is no anomaly for $\theta$ either 0 or $\pi$, but it is not possible to write continuous counterterms as a function of $\theta$ that preserve either $\mathsf{T}$ or $\mathcal{C}$ in the presence of background $A$ fields at both $\theta = 0, \pi$ [23]. (This situation was referred to in [38–40] as "a global inconsistency.") Again this implies that there must be a phase transition at some value of $\theta$ in agreement with our conclusion above.

Our conclusions agree with previous results and, significantly, extend them in new directions. Indeed, the focus of the previous analysis is on subtle aspects of discrete symmetries, while in the anomaly in the space of parameters (9) $\mathsf{T}$ and $\mathcal{C}$ play no role. This means that the anomaly in the space of parameters, and consequently our resulting dynamical conclusions, persists under $\mathsf{T}$ and $\mathcal{C}$-violating deformations. We illustrate this in a variety of systems below. This example is again characteristic. By isolating an anomaly in the space of parameters of a QFT we are able to see that the conclusions are robust under a large class of deformations, and apply to other theories.

## 1.4 Synthesis via Anomaly Inflow

We have now described two general physical problems of interest:

- Anomalies on the worldvolume of topologically non-trivial interfaces and defects created by position-dependent parameters.

- Discontinuous counterterms in a family of QFTs and consequences for the long-distance behavior.

One of the basic points of our analysis to follow is that the solution of these two conceptually distinct problems is unified via anomaly inflow. Indeed the same $(d + 1)$-dimensional classical theory can be used as a tool to analyze both phenomena. The difference between the applications is geometric. To describe a defect, the coupling constants vary in the physical spacetime. To describe a family of QFTs the coupling constants vary in the ambient directions extending the physical spacetime.

To illustrate this essential point, let us return again to the example of two-dimensional $U(1)$ gauge theory coupled to $N$ scalars. The same anomaly action (9) introduced to restore the $2\pi$ periodicity of $\theta$ in the presence of a background $A$ field can be used to compute the worldvolume anomaly of an interface, where $\theta$ varies smoothly from 0 to $2\pi n$, for integer $n$. In the first application the two spacetime dimensions have nonzero $w_2(A)$ and in the second, they have nonzero $d\theta$. In the latter case we simply integrate the anomaly to find

$$\omega_{\text{interface}} = \frac{n}{N} w_2(A) \,. \tag{10}$$

Since the bulk physics is trivially gapped for generic $\theta$, we can interpret the above result as the anomaly of the effective quantum mechanical degrees of freedom localized on the interface. We deduce that the ground states of this quantum mechanics are degenerate and they form a projective representation of the $PSU(N)$ symmetry (i.e. a representation of $SU(N)$) with $N$-ality $n$.

Thus we see that these two distinct physical applications are synthesized via anomaly inflow in the space of coupling constants.

## 1.5   An Intuitive Interpretation in Terms of $-1$-Form Symmetries

Unlike ordinary 't Hooft anomalies, our anomalies are not associated with global symmetries of the system. They describe subtleties in the interplay between global symmetries and identifications in the parameter space. However, in some cases we can make our anomalies look more like ordinary anomalies in global symmetries. The examples with the periodicity of the $\theta$-parameter can be thought of, somewhat heuristically, as related to a "$-1$-form global symmetry."

The $\theta$-parameter is coupled to the instanton number. Borrowing intuition from string theory, we can view the instantons as $-1$-branes. More precisely, instantons are not branes. They are not well-defined excitations in spacetime. Yet, for many purposes they can be viewed as branes. Since they are at a point in spacetime, these are $-1$-branes. Extending this view of instantons, we can think of them as carrying $-1$-form charge. Clearly, this is an abuse of language – this charge is not a well-defined operator acting on a Hilbert space.

By analogy with ordinary charges, we can view $\theta$ as a background classical "gauge field" for this $-1$-form symmetry. Since it is circle valued, it can have transition functions where it jumps by $2\pi\mathbb{Z}$, but its "field-strength" $d\theta$ is single-valued. Then, all our anomaly expressions are similar to ordinary anomalies, except that they involve also these kinds of "gauge fields".

We should stress, however, that as far as we understand, this intuitive picture of the anomaly associated with $\theta$ cannot be extended to other situations where the topology of the parameter space is different. For example, we do not know how to do it for the examples in section 3.

## 1.6   Another Synthesis

We describe the situation in the following terms. Let $F$ be a (Wick-rotated) $d$-dimensional field theory. It is defined on smooth manifolds $X$ endowed with extra structure—background fields—which may include both continuous fields, such as a Riemannian metric or a connection, and discrete fields, such as a spin structure or a "finite gauge field". Let $\mathcal{X}$ be the "parameter space"[11] for manifolds with this structure. We say that $F$ is a *field theory with domain* $\mathcal{X}$. Our main observation is that if we choose $\mathcal{X}$ suitably large, then typically there is an ('t Hooft) anomaly $\alpha$ which is an important and informative invariant of the theory $F$. Structurally, $\alpha$ is[12] an invertible $(d+1)$-dimensional field theory with the same domain $\mathcal{X}$, and $F$ is a theory *relative* to $\alpha$ in the sense of [41]. For example, if $X$ is a closed $d$-dimensional manifold equipped with background fields, then $\alpha(X)$ is a 1-dimensional complex vector space—a line—and the partition function $F(X)$ is an element of $\alpha(X)$; a similar statement holds for correlation functions. (By contrast, in an *absolute* field theory the partition function and correlation functions are complex numbers, i.e., elements of the trivial complex line.)

*Remark* 11. As a concrete example, consider the $\theta$-term in the two-dimensional gauge theory discussed in section 1.3. Before introducing the background PSU($N$) gauge field $A$ the (exponentiated) $\theta$-term is a well-defined complex-valued function. To extend the theory to include $A$ we must extend the definition of the $\theta$-term. That extension naturally takes values in a complex line—a 1-dimensional complex vector space with no natural basis—and so the extended theory is defined as an ('t Hooft) anomalous theory. We elaborate in Example 15 below. In general, the extension of an absolute theory with domain $\mathcal{X}'$ to a larger domain $\mathcal{X} \supset \mathcal{X}'$ may be a relative theory. The anomaly of that larger relative theory is part of its definition, not a computation from the smaller theory on $\mathcal{X}'$.

---

[11]$\mathcal{X}$ is not a topological space; see section 6.1 for an indication of what $\mathcal{X}$ is. A "point" of $\mathcal{X}$ is a manifold together with background fields. Our choice of notation in (16) and throughout is better suited to invertible field theories than noninvertible field theories, but the general idea pertains to all field theories.

[12]There are exceptions in which $\alpha$ is only defined as a field theory truncated to dimension $\leqslant d$.

*Remark* 12. There is a homotopy-theoretic framework for invertible field theories in which the isomorphism class of $\alpha$ is a generalized *differential* cohomology class on $\mathcal{X}$. (We give an introduction to differential cohomology in section 5.) This *isomorphism class* is the relevant invariant. If we deform the theory, for example by the renormalization group flow, then the isomorphism class of $\alpha$ may move but its *deformation class* is unchanged. The deformation class of $\alpha$ is a generalized *topological*[13] cohomology class on $\mathcal{X}$.

*Remark* 13. The dynamical argument with anomalies—'t Hooft anomaly matching—is based on the premise that an effective theory $F'$ has the same anomaly $\alpha$ as the original theory $F$, at least up to deformation. As stated at the end of section 1.1, the anomaly may change continuously under renormalization group flow, but its deformation class is unchanged, and this suffices to draw physical conclusions, as indicated at the end of section 1.3. For example, if $F$ is gapped with a single vacuum, then one expects the low energy effective field theory $F'$ to be invertible. But if the deformation class of the anomaly $\alpha$ is nontrivial, then no such $F'$ exists. We use this argument several times to show that for certain families of theories there must exist a phase transition or other interesting dynamical effect.

*Remark* 14. An anomaly is *not* an obstruction to constructing a "sensible" theory if we keep the fields in $\mathcal{X}$ nondynamical. On the other hand, an anomaly *is* an obstruction to making background fields dynamical. In this paper we only consider the former situation, in which we often say the theory has an *'t Hooft anomaly* to avoid the negative connotations of the naked term 'anomaly'. We give a more precise statement in Remark 20 below.

**Example 15.** To illustrate these ideas, in particular the domain of a field theory, we briefly discuss two-dimensional $U(1)$ gauge theory coupled to $N$ scalar fields of unit charge. This theory and its anomaly were introduced in section 1.3; see section 4.3 for more details.

First, the theory with constant $\theta$-parameter has domain $\mathcal{X}_1$ which is the total space of a vector bundle

$$\mathbb{C}^N \longrightarrow \mathcal{X}_1 \longrightarrow \mathcal{M}SO_{\text{Riem}} \times B_\nabla U(1), \tag{16}$$

where $\mathcal{M}SO_{\text{Riem}}$ classifies[14] oriented Riemannian manifolds and $B_\nabla U(1)$ classifies principal $U(1)$-bundles with connections ($U(1)$ gauge fields). A "point" of $\mathcal{X}_1$ is a smooth oriented Riemannian manifold $X$ equipped with a principal $U(1)$-bundle $P \to X$ with connection and a section of the associated rank $N$ vector bundle. $\mathcal{X}_1$ is the domain of the semiclassical gauge theory, an invertible field theory with partition function the exponential of (104). The $\theta$-term on a closed 2-manifold has the form

$$\exp(i\theta \deg(P)), \tag{17}$$

where $\deg(P)$ is the degree of the $U(1)$-bundle, the integral of its first Chern class. This semiclassical theory is absolute: there is no 't Hooft anomaly. We remark that to construct the quantum theory one integrates over the fibers of $\mathcal{X}_1 \to \mathcal{M}SO_{\text{Riem}}$, that is, one integrates out the $U(1)$ gauge field and the scalar fields.

The theory which promotes $\theta$ to a scalar field has domain $\mathcal{X}_2$ which is the total space of a vector bundle

$$\mathbb{C}^N \longrightarrow \mathcal{X}_2 \longrightarrow \mathcal{M}SO_{\text{Riem}} \times B_\nabla U(1) \times \mathbb{R}/2\pi\mathbb{Z}. \tag{18}$$

Each constant value of $\theta$ determines an embedding $\mathcal{X}_1 \to \mathcal{X}_2$. There is a non-anomalous, or absolute, extension of the $\theta$-term (17) to $\mathcal{X}_2$. It now depends on the $U(1)$ connection, not

---

[13]as opposed to differential

[14]The notation is geared to the invertible case, in which $\mathcal{M}SO_{\text{Riem}}$ encodes oriented Riemannian manifolds of all dimension. For the noninvertible case we use bordism categories and the dimension of the theory appears explicitly. We remark that the product notation in (16) and beyond is only appropriate in the invertible case. In this heuristic treatment we also ignore basepoints in the classifying objects.

just on the underlying bundle. Its definition uses the product of this connection and $\theta$ in differential cohomology. (A description in terms of Čech theory is given in section 4.1.1; see section 5.3 for an introduction to the general product in differential cohomology.)

As explained in section 4.3 the theory has a symmetry group $\mathrm{PSU}(N)$. The symmetry is implemented by extending the domain to a "space" $\mathfrak{X}_3$ which includes a background $\mathrm{PSU}(N)$ gauge field; it fits into a fibering

$$\mathbb{C}^N \times B_\nabla \mathrm{U}(1) \longrightarrow \mathfrak{X}_3 \xrightarrow{\ \pi\ } \mathcal{M}\mathrm{SO}_{\mathrm{Riem}} \times B_\nabla \mathrm{PSU}(N) \times \mathbb{R}/2\pi\mathbb{Z}. \tag{19}$$

The $\mathrm{U}(1)$ gauge field in the fiber and $\mathrm{PSU}(N)$ gauge field in the base combine to a $\mathrm{U}(N)$ gauge field. There is an embedding $\mathfrak{X}_2 \to \mathfrak{X}_3$ which lands on the trivial $\mathrm{PSU}(N)$ gauge field. The only extension we know of the $\theta$-term from $\mathfrak{X}_2$ to $\mathfrak{X}_3$ is anomalous: it takes values in a complex line rather than the complex numbers. Namely, (17) depends on the first Chern class of a $\mathrm{U}(1)$-bundle $P$, whereas on $\mathfrak{X}_3$ we only have a $\mathrm{U}(N)$-bundle $Q$. A trivialization of the underlying $\mathrm{PSU}(N)$-bundle reduces $Q$ to a $\mathrm{U}(1)$-bundle $P$, in which case $c_1(P) = c_1(Q)/N$. Hence the expression of the $\theta$-term in terms of $Q$ uses division by $N$. Since the first Chern class of a general $\mathrm{U}(N)$-bundle is not divisible by $N$, we cannot execute this division in integral cohomology. A consequence is that after exponentiation, as in (17), the $\theta$-term can be defined in a complex line rather than in the complex numbers (See [Paper 2, §3.1] for details in a similar example.). In other words, the extension of the $\theta$-term to $\mathfrak{X}_3$ that we use has an 't Hooft anomaly, and furthermore it is not even defined without first specifying the anomaly. (We cannot give an element of a vector space without first specifying the vector space.) The degree three cohomology class which defines the three-dimensional anomaly theory is depicted in (9).

*Remark* 20. The anomaly theory only depends on $\theta$ and the $\mathrm{PSU}(N)$-bundle, so as a theory with domain $\mathfrak{X}_3$ the anomaly is pulled back along $\pi$ from a theory on the base of the fibering (19). This means that there is no formal obstruction[15] to integrating along the fibers of $\pi$, that is, to integrating out the $\mathrm{U}(1)$ gauge field and the scalar fields. This is the precise sense in which an 't Hooft anomaly is not an obstruction to quantization. The quantum theory—with background fields a metric, orientation, $\mathrm{PSU}(N)$ gauge field, and scalar field $\theta$—has the same 't Hooft anomaly as the semiclassical theory.

We use two general scenarios to extract an absolute theory from a theory $F$ with ('t Hooft) anomaly $\alpha$. Scenario One does not involve more data, but it brings in $(d+1)$-dimensional manifolds. It is used repeatedly in subsequent sections of this paper and was already introduced in section 1.1. In this scenario we consider the coupled $(d+1)$-dimensional theory $(\alpha, F)$ in which $F$ is a *boundary theory* for $\alpha$. So if $Y$ is a compact $(d+1)$-dimensional manifold[16] with boundary $\partial Y = X$, and $X$ is "colored" with the boundary theory $F$, then $(Y, X)$ behaves as if it has no boundary, hence $(\alpha, F)$ evaluates to a complex number, as in (1.3). We remark that the relative $d$-dimensional theory is embedded in this coupled $(d+1)$-dimensional theory. Namely, to evaluate the relative theory on a closed $d$-dimensional manifold $X$, form $Y = [0,1] \times X$, color $\{0\} \times X$ with the boundary theory $F$, and leave $\{1\} \times X$ uncolored. The coupled theory evaluates on this manifold to the element $F(X)$ in the line $\alpha(X)$.

*Remark* 21. We obtain close cousins to $(\alpha, F)$ by two possible modifications: (1) tensor $F$ with an invertible $d$-dimensional field theory (with domain $\mathfrak{X}$), and (2) tensor $\alpha$ with an invertible $(d+1)$-dimensional theory whose truncation to $d$ dimensions is equipped with a "nonflat trivialization". (We discuss this last notion, a trivialization of the underlying topological theory,[17]

---

[15]On the level of partition functions and correlation functions, a usual anomaly is an obstruction since one cannot sum elements in distinct vector spaces. In this situation, since the anomaly theory is pulled back along $\pi$, the sum along the fibers of $\pi$ is a sum of elements in the same vector space.

[16]The background fields are implicit.

[17]Analogy: let $L \to M$ be a line bundle with covariant derivative over a smooth manifold. A *trivialization* is a nonzero flat section whereas a *nonflat trivialization* is a nonzero section whose covariant derivative is unconstrained; see section 5.2.

at the end of section 6.1.)

Scenario Two does not introduce $(d + 1)$-dimensional manifolds, but does bring in additional data. It is the *pullback* to a space of background fields on which the anomaly $\alpha$ is topologically trivialized. The data $(\phi, \tau)$ is a map

$$\phi: \mathcal{X}' \longrightarrow \mathcal{X} \tag{22}$$

and a nonflat trivialization $\tau$ of the pullback anomaly theory $\phi^*\alpha$. In this way we obtain an absolute theory with domain $\mathcal{X}'$.

*Remark* 23. The group of invertible $d$-dimensional theories with domain $\mathcal{X}'$ acts on the possible nonflat trivializations of $\phi^*\alpha$, and the orbit of the theory $(\phi^*F, \phi^*\alpha, \tau)$ consists of closely related theories. This is the Scenario Two parallel to Remark 21 for Scenario One. Elsewhere we call multiplication by an invertible $d$-dimensional theory the addition of "counterterms for the background fields". Notice that this action does not change the anomaly theory $\alpha$.

**Example 24.** We resume the discussion of Example 15 to illustrate Scenario Two. Recall that the theory with domain $\mathcal{X}_3$ has an anomaly. We have already indicated that its pullbacks along $\mathcal{X}_2 \to \mathcal{X}_3$ and $\mathcal{X}_1 \to \mathcal{X}_3$ are equipped with a trivialization of the pullback anomaly. A different pullback is indicated in section 4.3 in the discussion of extended periodicity. Namely, if we lift $\theta$, which has values in $\mathbb{R}/2\pi\mathbb{Z}$, to a scalar field with values in $\mathbb{R}/2\pi N\mathbb{Z}$, then there is a natural trivialization of the anomaly. The domain $\mathcal{X}'$ of the pullback theory fits into a fibering as in (19) in which $\mathbb{R}/2\pi\mathbb{Z}$ is replaced by $\mathbb{R}/2\pi N\mathbb{Z}$.

*Remark* 25. If a theory with domain $\mathcal{X}'$ has a nonzero anomaly, it is sometimes useful to determine the anomaly on another domain $\mathcal{X}$ and the compute the anomaly on $\mathcal{X}'$ by pulling back along a map (22). We illustrate this technique in section 7.1 and section 7.2; see (184) and (193).

## 1.7 Examples and Summary

Let us now summarize the examples analyzed below. We begin in section 2 with the elementary example of a quantum particle moving on a circle. This system is exactly solvable and exhibits a mixed anomaly between the circle-valued parameter $\theta$ and the $U(1)$ global symmetry or its $\mathbb{Z}_N$ subgroup. This example also gives us an opportunity to illustrate the various subtleties that occur when we make parameters position dependent.

In section 3 we discuss theories of free fermions in various spacetime dimensions as a function of the fermion mass. We start with the pedagogical example of a complex fermion in quantum mechanics. For $3d$ fermions we discuss how the index theorem of [42] explaining the discontinuity in the Chern-Simons level in the low-energy theory as real masses are varied from $-\infty$ to $\infty$ encodes an anomaly. For $4d$ Weyl fermions we describe the anomaly involving the complex fermion mass and explain its relationship to previous work [5, 19, 20] on fermions with position-dependent masses.

In section 4 we study $2d$ $U(1)$ gauge theory coupled to charged scalar fields. This includes Coleman's original paper [43] and the $\mathbb{CP}^n$ non-linear sigma model as special cases. These theories have a circle-valued coupling $\theta$ and we describe the resulting anomaly in the space of parameters. Here we extend the recent results of [23, 36, 37], which focused on the charge conjugation ($\mathcal{C}$) symmetry of these models at $\theta = 0, \pi$ to variants of these theories without this symmetry.

In sections 5–7 we present a more mathematical perspective on these anomalies. Section 5 is a mathematical interlude on differential cohomology. We encounter differential cohomology when making global sense of secondary invariants on a $k$-manifold without choosing an auxiliary $(k+1)$-manifold. This applies to Chern-Simons invariants, the Wess-Zumino-Witten term,

and many other examples. Our introduction here is offered as a gateway into the literature. We remark that some of the earliest appearances of differential cohomology in physics, implicitly and explicitly, are [44, 45]; the mathematical theory is now highly developed; see [46–49] and the references therein.

In section 6 we consider differential cohomology not on a single manifold, but simultaneously on all manifolds of a fixed dimension (equipped with background fields). These classes live on an abstract "space" that classifies manifolds. In the application to physics such differential cohomology classes are isomorphism classes of invertible field theories. The underlying topological cohomology class is the deformation class of the invertible field theory, as studied in [50]. In our application here the invertible field theories are in dimension $(d + 1)$ and are anomaly theories of $d$-dimensional field theories.

Section 7 applies the differential cohomology ideas to some of the systems discussed in this paper. In particular, we give geometric explanations for the isomorphism class of the anomaly in theories with a $\theta$-term. For free spinor fields with variable mass we also propose a precise formula for the isomorphism class of the anomaly; our formula (220) applies in arbitrary dimension with arbitrary spinor content. As in the vast literature on anomalies for massless spinor fields, we apply the circle of ideas around the Atiyah-Singer index theory to compute the anomaly for theories with variable mass. The new twist is to use Quillen's superconnections to encode the mass. Some cases (section 7.3) are covered by theorems in the existing literature [51]; for the general case (section 7.4) we make a conjecture.

Finally, in a companion paper [52] we discuss applications of these ideas to four-dimensional gauge theories, including the anomaly in parameter space for Yang-Mills theory with general gauge groups as well as extensions to theories with matter.

## 2  A Particle on a Circle

In this section we begin our generalization of the notion of anomalies to families of quantum systems. We present a simple and well-known example of a particle on a circle. This theory has a coupling constant $\theta$ and, as we show, exhibits an anomaly in this parameter space.

The dynamical variable in the theory is a periodic coordinate $q \sim q + 2\pi$. The Euclidean action is:

$$\mathcal{S} = \frac{m}{2} \int d\tau \, \dot{q}^2 - \frac{i}{2\pi} \int d\tau \, \theta \dot{q} \, . \tag{26}$$

In the above, $\theta$ is a coupling constant. Since the integral of $\dot{q}$ is quantized, the effect of $\theta$ is to weight different winding sectors with a phase. So defined, the parameter $\theta$ appears to be an angular variable with $\theta \sim \theta + 2\pi$. For instance the partition function, $Z$, viewed as a function of $\theta$ obeys

$$Z[\theta + 2\pi] = Z[\theta] \, . \tag{27}$$

Our goal in the following analysis is to clarify the circumstances where this periodicity of $\theta$ is valid. A distinguished role is played by the global shift symmetry under which $q \to q + \chi$. We will see that the $2\pi$ periodicity of $\theta$ is subtle in two related ways:

- If we try to make $\theta$ a non-constant function on the circle, the shift symmetry of $q$ can be broken.

- In certain special correlation functions, related to adding background fields for the shift symmetry, the $2\pi$-periodicity of $\theta$ is broken.

We elucidate these points and then discuss their dynamical consequences. See section 7.1 for further discussion of this theory and its anomaly.

## 2.1 Spacetime Dependent Coupling $\theta$

Let us first attempt to promote the coupling constant $\theta$ to depend on Euclidean time (which we assume to be periodic). Thus, we wish to make sense of the functional integral with action (26), where now $\exp(i\theta(\tau))$ is a given function to $\mathbb{S}^1$. Note that unlike the variable $q(\tau)$ in (26) which is summed over, the function $\exp(i\theta(\tau))$ is a fixed classical variable.

Our first task is to clarify the meaning of the integral:

$$\exp\left(\frac{i}{2\pi}\int d\tau\ \theta(\tau)\dot{q}(\tau)\right)\ . \tag{28}$$

In general for a periodic variable such as $q$ or $\theta$, the derivative is always a single-valued function. Hence when $\theta$ was a constant the integral above was well-defined. However, when we allow $\theta$ to be position dependent it may wind in spacetime and the integral requires clarification.

A systematic way to proceed is to divide the spacetime circle into patches $U_i$, where $i = 1, \cdots n$ labels the patches, and each patch is an open interval. (For simplicity we assume that the patches only intersect sequentially so $U_i \cap U_j$ is non-empty only when $i = j \pm 1$. We also treat $i$ as a cyclic variable.)

In each patch we can choose a lift $\theta_i : U_i \to \mathbb{R}$. On the non-empty intersections $U_i \cap U_{i+1}$ the lifts are related as

$$\theta_i = \theta_{i+1} + 2\pi n_i\ , \qquad n_i \in \mathbb{Z}\ . \tag{29}$$

The collection of lifts and transition functions yields a well-defined function to the circle. However it is redundant. If we adjust the data as

$$\theta_i \to \theta_i + 2\pi m_i\ , \qquad n_i \to n_i + m_i - m_{i+1}\ , \tag{30}$$

we obtain the same function $\exp(i\theta(\tau))$.

We now define the integral (28) using the collection of lifts $\theta_i$. In each intersection $U_i \cap U_{i+1}$ we choose a point $\tau_i$. We then set

$$\exp\left(\frac{i}{2\pi}\int d\tau\ \theta(\tau)\dot{q}(\tau)\right) \equiv \exp\left(\frac{i}{2\pi}\sum_{i=1}^{n}\int_{\tau_{i-1}}^{\tau_i} d\tau\ \theta_i(\tau)\dot{q}(\tau) - i\sum_{i=1}^{n} n_i q(\tau_i)\right)\ . \tag{31}$$

It is straightforward to verify the following properties of this definition:

- If the points $\tau_i$ defining the limits of the definite integrals are changed, the answer is unmodified.

- If the lifts $\theta_i$ and transitions $n_i$ are redefined preserving the function $\exp(i\theta(\tau))$ as in (30), the integral is unmodified.

- If the patches are refined, i.e. a new patch is added, the integral is unmodified.

- If we change the choice of lift of $q$, for instance if we replace $q(\tau_i) \to q(\tau_i) + 2\pi$, the integral is unmodified.

- In the special case of constant $\theta$, it reduces to the obvious definition.

Thus, the prescription (31) allows us to explore this quantum mechanics in the presence of spacetime varying coupling constant $\theta$. More formally, the above may be viewed as a product in differential cohomology. See section 5 for an introduction and in particular section 5.3 for a global definition of the product in this case.

It is important to stress that although the definition (31) privileges the points $\tau_i$, there is no physical operator inserted at these points. Rather, (31) is merely a way to define an integral in the case where $\theta$ has non-trivial winding number.

More physically, the varying coupling constant $\theta(\tau)$ allows us to illustrate the general comments on interfaces from section 1.2. When studying a problem with varying coupling constant $\theta(\tau)$ the simplest situation is that $\theta$ varies smoothly. In this case, the long distance behavior is intrinsic to the theory. We can generalize these situations where $\theta$ varies discontinuously with a sharp jump at some point $\tau_*$, so $\theta(\tau_* - \epsilon) = \theta(\tau_* + \epsilon) + a$ for some constant $a$. Such a configuration is commonly referred to as a sharp interface or defect. In this case the dynamics are not universal, as any defect may be modified by dressing it by an operator $\mathcal{O}(\tau_*)$. What the definition (31) shows, is that in the special case where the discontinuity $a$ is $2\pi n$ for some integer $n$, and the operator $\mathcal{O}(\tau_*)$ is taken to be $\exp(-inq(\tau_*))$ then the defect is trivial.

We can now explore aspects of the physics of varying coupling constant. Of particular interest is the interplay with the global symmetries of the problem. Consider a background $\theta(\tau)$ with non-zero winding number

$$L = \frac{1}{2\pi} \oint \dot{\theta}(\tau) d\tau = \sum n_i \tag{32}$$

around the circle. We claim that in such a configuration the global shift symmetry is broken. To see this, we shift $q(\tau) \to q(\tau) + \chi$ where $\chi$ is a constant. We then have from (31)

$$\exp\left(\frac{i}{2\pi} \int d\tau \, \theta(\tau)\dot{q}(\tau)\right) \to \exp\left(\frac{i}{2\pi} \int d\tau \, \theta(\tau)\dot{q}(\tau) - iL\chi\right) . \tag{33}$$

Since the remainder of the action (26) is obviously invariant under this shift, we can promote the above to a transformation of the full partition function under shifting $q \to q + \chi$

$$Z[\theta(\tau)] \to \exp\left(-\frac{i}{2\pi} \int d\tau \, \dot{\theta}(\tau)\chi\right) Z[\theta(\tau)] . \tag{34}$$

Since the zero mode of $q$ is integrated over, the above in fact means that correlation functions vanish unless the insertions are chosen to cancel this transformation. Specifically, consider inserting $\prod_j \exp(i\ell_j q(\tau_j))$ together with other operators depending only on $\dot{q}$. From (34) we see that these correlation functions are non-zero only if $\sum_j \ell_j = L$.

As emphasized in section 1.1, equation (34) is characteristic of phenomena typically referred to as anomalies. By activating a topologically non-trivial configuration for the background field $\theta(\tau)$, in this case a non-zero winding number, the global $U(1)$ shift symmetry is violated.

## 2.2 Coupling to Background $U(1)$ Gauge Fields

Another approach to the same problem is to study the particle on the circle, to begin with at constant coupling $\theta$, in the presence of a background $U(1)$ gauge field $A = A_\tau d\tau$ for the global $U(1)$ shift symmetry. The action is modified to[18]

$$S = \frac{m}{2} \int d\tau (\dot{q} - A_\tau)^2 - \frac{i}{2\pi} \int d\tau \, \theta(\dot{q} - A_\tau) , \tag{35}$$

and is invariant under gauge transformations $q \to q + \Lambda(\tau)$ and $A_\tau \to A_\tau + \dot{\Lambda}(\tau)$. (Note that since $\Lambda(\tau)$ transforms the classical field $A_\tau$, it should be viewed as classical, and as such it cannot be used to set the dynamical degree of freedom $q$ to zero.) The path integral over

---

[18]See section 7.1 for an explication of (35) in which $q$ is a section of a principal $U(1)$-bundle.

$q$ now yields a partition function $Z[\theta,A]$ depending on a parameter $\theta$ and a gauge field $A$. However, it is no longer $2\pi$-periodic in $\theta$. Instead:

$$Z[\theta + 2\pi,A] = Z[\theta,A]\exp\left(-i\int d\tau A_\tau(\tau)\right)\,. \tag{36}$$

One possible reaction to the equation above is simply that the $2\pi$-periodicity of the parameter $\theta$ is incorrect. Instead, more precisely, when discussing the coupling to background gauge fields $A$ we should take care to also specify the counterterms, i.e. the local functions of the background fields that may be added to the action. In this case the counterterm of interest is a one-dimensional Chern-Simons term for the background gauge field $A$. Thus more precisely we can write the action

$$S = \frac{m}{2}\int d\tau(\dot{q} - A_\tau)^2 - \frac{i}{2\pi}\int d\tau\ \theta(\dot{q} - A_\tau) - ik\int d\tau A_\tau(\tau)\,, \tag{37}$$

where $k$ must be an integer to ensure gauge invariance. Including such a term in the action we arrive at a partition function $Z[\theta,k,A]$. Then (36) means that

$$Z[\theta + 2\pi,k,A] = Z[\theta,k-1,A]\,. \tag{38}$$

Thus, in this interpretation the true parameter space is a helix that we may view as a covering space of the circle (where $\theta$ ranges from 0 to $2\pi$). The different values of $k$ index the different sheets in the cover. Said differently, if we demand that two values of the coupling constant are only considered equivalent if all observables agree, including the phase of the partition function in the presence of background fields, then there is, by definition, no such thing as an anomaly in the space of coupling constants.

## 2.3 The Anomaly

There is however, an alternative point of view, which is also fruitful. Instead of enlarging the parameter space, we can retain the $2\pi$-periodicity of $\theta$ as follows. We pick a two-manifold $Y$ with boundary the physical quantum mechanics worldvolume of our problem. We extend the classical fields $\theta$ and $A$ into the bulk $Y$. Then, we define a new partition function as

$$\tilde{Z}[\theta,k,A] = Z[\theta,k,A]\exp\left(2\pi i\omega[\theta,A]\right) = Z[\theta,k,A]\exp\left(\frac{i}{2\pi}\int_Y \theta F\right)\,. \tag{39}$$

This modified partition function now obeys the simple $2\pi$-periodicity of $\theta$ even in the presence of background fields

$$\tilde{Z}[\theta + 2\pi,k,A] = \tilde{Z}[\theta,k,A]\,. \tag{40}$$

The price we have paid is that $\tilde{Z}$ now depends on the chosen extension of the classical fields into the bulk $Y$.

We can now easily extend our analysis to allow for a non-constant function $\theta(\tau)$. The integral over $Y$ is defined similarly to (31). We divide the manifold $Y$ into patches in each patch we integrate a lift of $\theta$ times the curvature $F$. We add to this integral a boundary contribution, which in this case is a line integral of the gauge field $A$ weighted by the transition function $\theta_i - \theta_j$. On a closed two-manifold this results in a well-defined action independent of trivialization: moving the boundary of a patch $U_1$ to encompass a new region $W$ formerly contained in $U_2$ leads to a new integral $\exp\left(\frac{i}{2\pi}\int_W(\theta_{U_1} - \theta_{U_2})F\right)$ together with a compensating contribution $\exp\left(-\frac{i}{2\pi}\int_{\partial W}(\theta_{U_1} - \theta_{U_2})A\right)$.

To see the interplay with the boundary action (31) consider now a patch $U$ that terminates on the boundary. The result is now $U(1)$ gauge invariant even in the presence of general

$\theta(\tau)$. Indeed, the Wilson lines on the edge of each bulk patch now terminate on the insertions of $\exp(-in_i q(\tau_i))$ and hence are gauge invariant. Thus, the result is a partition function $\tilde{Z}[\theta(\tau), A]$ that is a well-defined function of a circle-valued field $\exp(i\theta(\tau))$ and gauge invariant as a function of $A$. For more details on this integral, see the discussion in section 4.1.1 and section 5.

For some purposes it is also useful to apply the descent procedure again to produce an anomaly three-form. Such a two-step inflow is in general possible when discussing infinite order anomalies (classified by an integer) such as those computed by one-loop diagrams in even-dimensional QFTs. In this case we find:

$$\mathcal{I}_3 = d\omega = \frac{1}{(2\pi)^2} d\theta \wedge F \,, \tag{41}$$

which encodes the anomaly action in (39).[19] We will revisit this anomaly in section 7.1.

## 2.4 Dynamical Consequences: Level Crossing

We can use our improved understanding of the behavior of the theory as a function of the $\theta$-angle to make sharp dynamical predictions about the particle on a circle. Specifically, we claim that for at least one value $\theta_* \in [0, 2\pi)$, the system must have a non-unique ground state.

To argue for this, suppose on the contrary that we have a unique ground state for all $\theta$. We can focus on this state by scaling up all the energies in the problem. At each $\theta$, the low-energy partition function is then that of a trivial system with a single unique state. However, a single unique state cannot produce the jump (36) in the one-dimensional Chern-Simons level. Here it is crucial that the coefficient of this level is quantized. In particular, this prohibits a continuously variable phase of the partition function interpolating between the value at $\theta = 0$ and $\theta = 2\pi$.

Of course, the free quantum particle on the circle is an exactly solvable system for any $\theta$ and its behavior is well-known. There is a single unique ground state for all $\theta \neq \pi$. However for $\theta = \pi$, where there is an enhanced charge conjugation symmetry, $\mathcal{C}$, acting as $q \to -q$, there are two degenerate ground states. Thus, the conclusions above are indeed correct, though the highbrow reasoning is hardly necessary. In terms of anomalies one may derive the degeneracy at $\theta = \pi$, following [23], by noting that for this value of $\theta$, there is a mixed anomaly between $U(1)$ and $\mathcal{C}$ and hence a unique ground state is forbidden.

The advantage of the more abstract arguments is that they are robust under a large number of possible deformations of this system. It is instructive to proceed in steps. We can first consider deforming the system by a potential breaking $U(1)$ to $\mathbb{Z}_N$ and preserving the other discrete symmetries

$$\begin{aligned} S &= \int d\tau \left( \frac{m}{2}\dot{q}^2 - \frac{i}{2\pi}\theta\dot{q} + U(q) \right), \\ U(q) &= U(q + \frac{2\pi}{N}) = U(-q) \,, \end{aligned} \tag{42}$$

e.g. $U(q) = \cos(Nq)$. Here, $U(q) = U(q + \frac{2\pi}{N})$ guarantees the $\mathbb{Z}_N$ symmetry and $U(q) = U(-q)$ guarantees that the $\mathcal{C}$ and $\mathcal{T}$ symmetries are as in the problem without the potential. For even $N$, there is a mixed anomaly at $\theta = \pi$ between $\mathcal{C}$ and $\mathbb{Z}_N$ leading again to ground state

---

[19]We can also change the precise representative of the cohomology class appearing in (41) without modifying the essential consequences in (40). This means that we can replace $d\theta$ by $d(\theta + f(\theta))$ with $f(\theta)$ a $2\pi$ periodic function. In fact as we will see in section 3.1, the low-energy theory near $\theta = \pi$ produces a different anomaly action $\omega$ related by continuously varying the form $d\theta/2\pi$ to a periodic delta function $\delta(\theta - \pi)d\theta$.

degeneracy.[20] For odd $N$ there is no anomaly, but it is impossible to define continuous $\theta$-dependent counterterms to preserve $\mathcal{C}$ at both $\theta = 0$ and $\theta = \pi$ [23]. This lack of suitable counterterms was referred to as a global inconsistency in [38,40], also implies a level crossing. In this theory our anomaly in the space of couplings persists and yields the same conclusion though it does not single out $\theta = \pi$ as special.

Finally, we can consider deformations breaking all symmetries, and in particular $\mathcal{C}$ and $\mathcal{T}$, except the $\mathbb{Z}_N$ symmetry. For instance, we can introduce a real degree of freedom $X$ and consider the action:

$$
\begin{aligned}
\mathcal{S} &= \int d\tau \left( \frac{m}{2}\dot{q}^2 - \frac{i}{2\pi}\theta\dot{q} + U(q) + \frac{M}{2}\dot{X}^2 + iX\dot{q} + V(X) \right), \\
U(q) &= U(q + \frac{2\pi}{N}),
\end{aligned}
\tag{43}
$$

with generic $U(q)$ (subject to $\frac{2\pi}{N}$ periodicity) and $V(X)$. Note that unlike (42), we do not impose that $U(q)$ is even and therefore we break $\mathcal{C}$ and $\mathcal{T}$ for all $\theta$. The free particle on a circle is obtained in the limit $U(q) \to 0$ and $M \to \infty$. The condition $U(q) = U(q+\frac{2\pi}{N})$ guarantees that the $\mathbb{Z}_N$ symmetry remains, however there is no special value of $\theta$ with enhanced symmetry. Nevertheless as we vary $\theta$ from zero to $2\pi$ the phase of the partition function changes by the insertion of a $\mathbb{Z}_N$ Wilson line, and thus the anomaly in the space of couplings persists. Hence if $N > 1$ we again deduce that somewhere in $\theta$ we must have level crossing for the ground state and hence ground state degeneracy.

To deduce how the anomaly action in (39) reduces in this more complicated situation, it is helpful to integrate the action there by parts and express it as

$$
\mathcal{A}[\theta, A] = \exp\left( -i \int_Y \frac{d\theta}{2\pi} A \right).
\tag{44}
$$

On a closed two-manifold $Y$ this defines the same anomaly action. On a manifold $Y$ with boundary, (44) and (39) differ by a choice of boundary counterterm (in this case $\theta A$.) In the following, we mostly use expressions such as (44) with the understanding that we may need to add suitable boundary terms.

Using (44), we can describe the anomaly action in the deformed theory (43) more precisely as follows. The $U(1)$ background gauge field $A$ is now replaced by a $\mathbb{Z}_N$ gauge field $K$. (Our convention is that the holonomies of $K$ are integers modulo $N$.) Concretely, we can embed $K$ in a $U(1)$ gauge field $A$ as $A = \frac{2\pi}{N}K$. Then we find that (44) reduces to the anomaly action

$$
\mathcal{A}[\theta, K] = \exp\left( -\frac{2\pi i}{N} \int_Y \frac{d\theta}{2\pi} \cup K \right).
\tag{45}
$$

This anomaly is non-trivial and implies the level-crossing of that system.[21]

In fact even in the general system (43), we can give a straightforward argument for level crossing using a canonical quantization picture. The wavefunctions of states of definite charge $k \bmod N$ under the $\mathbb{Z}_N$ symmetry can be expanded in a Fourier series as

$$
\psi(q, X) = \sum_{j=-\infty}^{\infty} e^{i(k+Nj)q} \mu_j(X).
\tag{46}
$$

---

[20]The anomaly action is $\exp\left(i\pi \int C \cup K\right)$ where $C$ is the $\mathbb{Z}_2$ charge conjugation gauge field and $K$ is the $\mathbb{Z}_N$ gauge field. Note that this is only non-trivial when $N$ is even. (Although charge conjugation acts non-trivially on the $\mathbb{Z}_N$ symmetry with gauge field $K$, this action is trivial once we reduce modulo two.)

[21]One can also construct a direct analog of the $i\int_Y \theta F$ anomaly action even in the case of a discrete $\mathbb{Z}_N$ symmetry. To do so, we lift the $\mathbb{Z}_N$ gauge field to a $U(1)$ gauge field and evaluate the integral using the differential cohomology definition in section 4.1.1.

Table 1: Summary of anomalies and existence of continuous counterterms preserving $\mathcal{C}$ ("global inconsistency") in the hierarchy of theories considered above. The left-most column shows the theory and its symmetry at generic values of $\theta$. Without using the charge conjugation symmetry, all these theories exhibit a mixed 't Hooft anomaly in $\theta$ and $\mathcal{G}$. The anomaly implies that the theories cannot have a unique ground state for all values of $\theta \in [0, 2\pi)$. For the simpler theories there is also a charge conjugation symmetry at $\theta = \pi$, which may suffer from an 't Hooft anomaly. We have also indicated when the theories lack a continuous counterterm that can preserve $\mathcal{C}$ at both $\theta = 0, \pi$.

| theory | without $\mathcal{C}$ | with $\mathcal{C}$ at $\theta = 0, \pi$ | |
| --- | --- | --- | --- |
| generic symmetry $\mathcal{G}$ | $\theta$-$\mathcal{G}$ anomaly | $\mathcal{C}$-$\mathcal{G}$ anomaly at $\theta = \pi$ | no continuous counterterms |
| $q$ <br> $\mathcal{G} = U(1)$ | ✓ | ✓ | ✓ |
| $q$ + potential (42) <br> $\mathcal{G} = \mathbb{Z}_N$ , $N > 1$ | ✓ | even $N$ ✓ <br> odd $N$ ✗ | even $N$ ✓ <br> odd $N$ ✓ |
| $q$ + $X$ system (43) <br> $\mathcal{G} = \mathbb{Z}_N$ , $N > 1$ | ✓ | No $\mathcal{C}$ symmetry | No $\mathcal{C}$ symmetry |

Let $\psi$ above be the ground state at $\theta = 0$. We can track this state as a function of $\theta$. The Hamiltonian is

$$H = \frac{1}{2m}\left(P_q - \frac{\theta}{2\pi} - X\right)^2 + \frac{1}{2M}P_X^2 + V(X) + U(q) \,. \tag{47}$$

In canonical quantization, the momentum operator is $P_q = -i\frac{d}{dq}$. From this we see that the ground state at $\theta = 2\pi$ is not $\psi(q, X)$, but rather is $e^{iq}\psi(q, X)$. Physically, this means that as we dial $\theta$ from zero to $2\pi$, the $\mathbb{Z}_N$ charge of the ground state wavefunction increases by one unit. In particular, at some value of $\theta$, level crossing for the ground state must occur.

## 3 Massive Fermions

In this section we consider free fermions in various spacetime dimensions as a function of their mass parameters. We will see that this gives simple examples of systems with anomalies in their parameter space. We will also see how these models can be deformed to interacting theories with the same anomaly. We discuss free fermions from a different viewpoint in sections 7.3–7.4.

### 3.1 Fermion Quantum Mechanics

Consider the quantum mechanics of a complex fermion with a real mass $m$. Anomalies in fermionic quantum mechanics were first discussed in [53]. The Euclidean action is

$$\mathcal{S} = \int d\tau \left(i\psi^\dagger \partial \psi + m\psi^\dagger \psi\right) \,. \tag{48}$$

This theory has a two-dimensional Hilbert space spanned by two energy eigenstates $|\pm\rangle$ of energy $E = \pm m/2$. On a Euclidean time circle of length $\beta$ the partition function is

$$Z[m] = e^{-\beta m/2} + e^{\beta m/2} \,. \tag{49}$$

At vanishing mass $m$ the theory has two degenerate ground states, while for non-zero mass, one or the other state becomes energetically favorable. As we will see, this means that this

fermion quantum mechanics is identical to the theory of a particle on a circle described in section 2 where we have isolated the two nearly degenerate states at $\theta = \pi$. (See e.g. [54].)

Of particular interest to us is the asymptotic behavior of the theory for large $|m|$. Regardless of the sign of $m$ we see that in this limit there is a single ground state and an infinite energy gap to the next state. Thus, the physics in these two limits is identical. Effectively we can say that the parameter space of masses is compactified to $\mathbb{S}^1$.

This simple free fermion theory has a $U(1)$ global symmetry and can be coupled to a background gauge field $A = A_\tau d\tau$, which modifies the action to

$$\mathcal{S} = \int d\tau \left( i\psi^\dagger(\partial - iA_\tau)\psi + m\psi^\dagger\psi - ikA_\tau \right) . \tag{50}$$

Here we have included in the action a counterterm depending only on $A$, whose coefficient $k$ is integral. Since we transition between the two states by an action of the $\psi$ operator, they differ in their $U(1)$ charge by one unit. Therefore the partition function including $A = A_\tau d\tau$ is (below we have shifted the Hamiltonian so that the energies are $\pm m/2$)

$$Z[m,k,A] = e^{ik\oint A} \left( e^{-\beta m/2} + e^{\beta m/2 - i\oint A} \right) . \tag{51}$$

Now we see that the theories at large positive and negative mass differ by a local counterterm

$$\lim_{m\to+\infty} \frac{Z[m,k,A]}{Z[-m,k,A]} = \exp\left( -i\oint A \right) . \tag{52}$$

Note crucially that since $k$ is quantized, there is no way to modify the result (52) by adding a continuous $m$-dependent counterterm for the background gauge field $A$. This means that we can interpret (52) as an anomaly involving the mass parameter and the $U(1)$ global symmetry. Specifically, we define a new partition function by extending the backgrounds, in this case the gauge field $A$ and the mass $m$, into a $2d$ bulk $Y$. We then define a new partition function by

$$\tilde{Z}[m,k,A] = Z[m,k,A] \exp\left( i\int \rho(m)F \right) , \tag{53}$$

where $F = dA$ is the curvature and the function $\rho(m)$ satisfies

$$\lim_{m\to-\infty} \rho(m) = 0 , \qquad \lim_{m\to+\infty} \rho(m) = 1 . \tag{54}$$

The modified partition function $\tilde{Z}$ now has a manifestly uniform limit as $|m|$ becomes large:

$$\lim_{m\to+\infty} \frac{\tilde{Z}[m,k,A]}{\tilde{Z}[-m,k,A]} = 1 . \tag{55}$$

This anomaly persists under arbitrary deformations of the theory that preserve the $U(1)$ symmetry. (For instance it persists under deformations that violate the charge conjugation symmetry, $\mathcal{C}$, which acts as $\mathcal{C}(\psi) = \psi^\dagger$.)

How shall we interpret the arbitrary function $\rho(m)$ above? One way to understand the ambiguity in the function $\rho(m)$ is that it reflects the fact that in general systems without additional symmetry there is no preferred way to parameterize the space of masses. Under a redefinition $m \to h(m)$ where $h(m)$ is any bijective function with $h(\pm\infty) = \pm\infty$ modifies the precise function $\rho$ above but preserves the properties (54). This is similar to the general counterterm ambiguities that are always present when discussing anomalies, and in fact parameter redefinitions occur along renormalization group trajectories.

It is the rigid limiting behavior of the function $\rho(m)$ above that means that the deformation class of the anomaly we are describing is preserved under any continuous deformation of the

theory. As in section 2.3, we can make the cohomological properties more manifest by applying the descent procedure again to produce an anomaly three-form (see footnote 6). In this case we find:

$$\mathcal{I}_3 = \frac{1}{2\pi} f(m) dm \wedge F \,, \tag{56}$$

where $f(m)dm$ is a one-form with the property that

$$\int_{-\infty}^{+\infty} f(m)dm = 1 \,. \tag{57}$$

In this free fermion problem, it is natural to take $f(m) = \delta(m)$, such that the anomaly is supported only at $m = 0$ where we have level crossing. This is in accord of the discussion in footnote 19. Below we will also discuss other options. Such a one-form represents a non-trivial cohomology class on the real line, once one imposes a decay condition for $|m| \to \infty$. (Here we have in mind a model such as compactly supported cohomology see e.g. [55]). The form $f(m)dm$ is non-trivial because it cannot be expressed as the derivative of any function tending to zero at $m = \pm\infty$. Alternatively as suggested above, one can compactify the real mass line to a circle in which case $f(m)dm$ represents the generator of $H^1(\mathbb{S}^1, \mathbb{Z})$. Viewed as such a cohomology class the anomaly is rigid because the integral is quantized. This feature is preserved under any continuous deformation of the theory.

## 3.2 Real Fermions in $3d$

As our first example of a quantum field theory (as opposed to a quantum mechanical theory) with an anomaly in parameter space we consider free fermions in three dimensions. We will see how some familiar properties of fermion path integrals can be reinterpreted as anomalies involving the fermion mass. We focus on the theory of a single Majorana fermion $\psi$, though our analysis admits straightforward extensions to fermions in general representations of global symmetry groups.

The Euclidean action of interest is

$$\mathcal{S} = \int d^3x \, (i\psi\slashed{\partial}\psi + im\psi\psi) \,, \tag{58}$$

where $m \in \mathbb{R}$ above is the real mass. Our goal is to understand properties of the theory as a function of the mass $m$. As above it is fruitful to encode these in a partition function $Z[m]$.

As in our earlier examples, we first consider the free fermion theory in flat spacetime. At non-zero $m$, the theory is gapped with a unique ground state and no long range topological degrees of freedom. As the mass is increased the gap above the ground state also increases and we isolate the ground state. In particular the partition function, as well as the correlation functions of all local operators, become trivial in this limit[22]

$$\lim_{m \to -\infty} Z[m] = \lim_{m \to +\infty} Z[m] = 1 \,. \tag{59}$$

Like the fermion quantum mechanics problem of section 3.1, one can interpret the above in terms of the effective topology of the parameter space. The space of masses is a real line, and we can formally compactify it to $\mathbb{S}^1$ by including $m = \infty$.

The situation is more subtle if we consider the theory on a general manifold with non-trivial metric $g$, and hence associated partition function $Z[m,g]$. Fixing $g$ but scaling up the mass again leads to a trivially gapped theory, however now the theories at large positive and

---

[22]The partition function $Z[m]$ is subject to an ambiguity by adding counterterms of the form $\int d^3x\, h(m)$ for any function $h(m)$. Below we assume that these terms are tuned so that (59) is true.

negative mass differ in the phase of the partition function. Locality implies that the ratio of the two partition functions in this limit must be a well-defined classical functional of the background fields. In this case the APS index theorem [42] implies that the ratio is[23]

$$\lim_{m \to +\infty} \frac{Z[+m, g]}{Z[-m, g]} = \exp\left(i \int_X CS_{\text{grav}}\right) , \tag{60}$$

where $CS_{\text{grav}}$ is the minimally consistent spin gravitational Chern-Simons term for the background metric.[24] Thus in the presence of a background metric, the identification between $m = \pm\infty$ is broken. (For early discussion of this in the physics literature, see [57, 58].)

Notice that in (60) we have focused only on the ratio between the partition functions. In fact since the theories at large $|m|$ are separately trivially gapped, each theory separately gives rise to a local effective action of the background metric. However in general, one may adjust the UV definition of the theory by adding such a local action for the background fields. Physically this is the ambiguity in adjusting the regularization scheme and counterterms. By considering the ratio of partition functions we remove this ambiguity. Thus while the effective gravitational Chern-Simons level is individually scheme-dependent for large positive and large negative mass, the difference between the levels is physical. (See also footnote 23.)

In fact, the jump in the gravitational Chern-Simons level (60) is a manifestation of the time-reversal (T) anomaly of the free fermion theory. At vanishing mass the system is time-reversal invariant, but the mass breaks this symmetry explicitly with $T(m) = -m$. The gravitational Chern-Simons term is also odd under T and hence a fully time-reversal invariant quantization of the theory in the presence of a background metric would require the effective levels at large positive and negative masses to be opposite. The jump formula (60) means that this is impossible to achieve by adjusting the counterterm ambiguity.

We would now like to reinterpret the jump (60) in terms of an anomaly involving the fermion mass viewed now as a background field. Analogous to our examples in quantum mechanics, we introduce a new partition function $\tilde{Z}[m, g]$, which depends on an extension of the mass $m$ and metric $g$ into a four-manifold $Y$ with boundary $X$:

$$\tilde{Z}[m, g] = Z[m, g] \exp\left(-i \int_Y \rho(m) dCS_{\text{grav}}\right) = Z[m, g] \exp\left(-\frac{i}{192\pi} \int_Y \rho(m) \text{Tr}(R \wedge R)\right) , \tag{62}$$

where above $\rho(m)$ satisfies the same criteria as in the anomaly in the fermion quantum mechanics theory (54). (And as in the discussion there, in the free fermion theory it is natural to take $\rho(m)$ a Heaviside theta-function.) This partition function now retains the identification between $m = \pm\infty$ even in the presence of a background metric $g$ at the expense of the extension into four dimensions.

In fact, using time-reversal symmetry we can say more about the function $\rho(m)$ above. Since $m$ is odd under T and time-reversal changes the orientation of spacetime, demanding that the 4d anomaly action in (62) is T invariant leads to the additional constraint

$$\rho(m) + \rho(-m) = 1 . \tag{63}$$

---

[23]As in footnote 22, below we use the freedom to tune counterterms. However, as we discuss the right-hand side of (60) cannot be modified by any such tuning.

[24]As usual, it is convenient to define this term by an extension to a spin four-manifold $Y$. Then for any integer $k$ we have

$$\exp\left(ik \int_X CS_{\text{grav}}\right) = \exp\left(2\pi ik \int_Y \frac{p_1(Y)}{48}\right) = \exp\left(\frac{ik}{192\pi} \int_Y \text{Tr}(R \wedge R)\right) , \tag{61}$$

where $p_1(Y)$ is the Pontrjagin class and we have used $\int_Y p_1(Y) \in 48\mathbb{Z}$ for any closed spin manifold $Y$. Although this term is called a gravitational 'Chern-Simons term' in the physics literature, it is not covered by the work of Chern-Simons [56]. Rather, it is an exponentiated $\eta$-invariant; see Remark 176.

In particular we can use this to recover the T anomaly of the theory at $m = 0$: using $\rho(0) = 1/2$, the anomaly becomes a familiar gravitational $\theta_g$-angle at the non-trivial T-invariant value of $\theta_g = \pi$.

However, the virtue of viewing the anomaly (62) as depending only on the parameters $m$ and $g$ is that it is manifestly robust under T violating deformations. This means that the anomaly (62) has implications for a much broader class of theories. For example, consider coupling the free fermion to a real scalar field $\varphi$ so the action is now

$$S = \int d^3x \left( i\psi\slashed{\partial}\psi + (\partial\varphi)^2 + i(m + \varphi)\psi\psi + V(\varphi) \right) , \tag{64}$$

where $V(\varphi)$ is any potential. For generic $V(\varphi)$ this system does not have T symmetry. Nevertheless the arguments leading to the anomaly involving the fermion mass $m$ and the metric $g$ still apply. In this more general context, the constraint (63) does not hold, and only the general constraint (54) is applicable.

As in section 2.3, we can also apply the descent procedure again to find an anomaly five-form:

$$\mathcal{I}_5 = -\frac{1}{384\pi^2} f(m)dm \wedge \mathrm{Tr}(R \wedge R) = -\frac{1}{48} f(m)dm \wedge p_1 , \tag{65}$$

where $p_1$ is the first Pontrjagin class of the manifold, and $f(m)dm = d\rho(m)$ has unit total integral. For the free spinor field we compute a particular $f(m)$ in section 7.4 using geometric index theory; see (226).

### 3.2.1 Dynamical Consequences

We now apply the anomaly (65) to extract general lessons about the physics. As described in section 1, there are broadly speaking two lessons that we can learn.

- Existence of non-trivial vacuum structure: Consider any QFT with the anomaly (65). Such a theory cannot have a trivially gapped vacuum (i.e. a unique ground state and an energy gap with no long-range topological degrees of freedom) for all values of the mass $m$. To argue for this we assume on the contrary that the theory is trivially gapped for all $m$. Then at long-distances $Z[m] \to 1$ for all masses $m$, which of course does not have the anomalous transformation required by the bulk anomaly action.

  Thus, we conclude that somewhere in the space of mass parameters the vacuum must be non-trivial. In other words, either the gap must close or a first order phase transition (leading to degenerate vacua) occurs. Of course for the free fermion this is hardly surprising since at $m = 0$ the fermion is massless. However for more general interacting systems such as that in (64), this conclusion is less immediate.

- Non-trivial physics on interfaces: Consider for instance a situation where for sufficiently large $|m|$ the theory is gapped. We activate a smooth space-dependent mass $m(x)$ depending on a single coordinate $x$ which obeys $m(\pm\infty) = \pm\infty$. At low-energies in the transverse space we find an effective theory, which is necessarily non-trivial. The anomaly of this theory can be computed by integrating the anomaly action over the coordinate $x$. Using the property (57) this leads to

$$i \int_{Y_3} CS_{\mathrm{grav}} , \tag{66}$$

  where now $Y_3$ is an extension of the effective two-dimensional theory. In particular, the anomaly (66) implies that the theory on the interface is gapless with chiral central charge fixed by the anomaly theory $c_L - c_R = 1/2$. This result is well-known in the condensed

matter literature: the classical action (66) describes a 3$d$ topological superconductor without a global symmetry, which is known to have a gapless chiral edge mode.

Again for the free fermion this conclusion is obvious. At the special locus in $x$ where $m = 0$, the 3$d$ fermion becomes localized and leads to a massless 2$d$ Majorana-Weyl fermion. However, for more general interacting systems with the same anomaly, the conclusion still holds.

In general, the basic idea encapsulated by the above example is that for any one-parameter family of generically gapped systems with symmetry $\mathcal{G}$ (either unitary internal or spacetime) we can track the long-distance $\mathcal{G}$ counterterms as a function of the parameter. The discontinuity in these counterterms as the parameter is varied from $-\infty$ to $\infty$ is an invariant of the family.

Such tracking of the jump in gravitational and other Chern-Simons terms in background gauge fields was a powerful consistency check on various conjectures about 3$d$ dynamics and dualities [24–35]. Here, we see that this idea is formalized into an anomaly in the space of coupling constants and this consistency check is unified with standard anomaly matching.

### 3.3   Weyl Fermions in 4$d$

We now consider free fermions in 4$d$. We will again find mixed anomalies in the space of mass parameters and background metrics. A qualitatively new feature is that in this case the anomaly is present only if we study the full two-dimensional complex $m$-plane. Effectively, this means that the $m$-plane is a non-trivial two-cycle in parameter space. Other examples with two-cycles in parameter space are discussed in [9,10,21]. We focus below on the simplest case of a minimal free Weyl fermion. Extensions to fermions in general representations of global symmetry groups are straightforward.

Our starting point is the partition function $Z[m]$ of a free Weyl fermion $\psi$ viewed as a function of the complex mass parameter $m$. The massless theory has a chiral $U(1)$ symmetry under which $\psi$ has unit charge. A non-zero mass parameter entering the Lagrangian as $m\psi\psi + c.c$ breaks this symmetry and we can view $m$ as a spurion of charge minus two. This means that the partition function in flat space obeys (with an appropriate choice of counterterms):

$$Z\left[e^{i\phi}m\right] = Z\left[m\right] . \tag{67}$$

In particular, the above equation holds for large $|m|$ where the theory is trivially gapped. Thus it is consistent to compactify the mass parameter space to a sphere $\mathbb{S}^2$ by viewing all masses of large absolute value as a single point.

We now couple the theory to a background metric $g$ and reexamine the above conclusions. As in our example in 3$d$, we will see that the large $|m|$ behavior of the partition function is now more subtle. Recalling that for $m = 0$ the $U(1)$ chiral symmetry participates in a mixed anomaly with the geometry, the partition function is modified under a chiral rotation as:

$$Z\left[e^{i\phi}m, g\right] = Z\left[m, g\right]\exp\left(-\frac{i\phi}{384\pi^2}\int_X \text{Tr}(R \wedge R)\right) = Z\left[m, g\right]\exp\left(-i\phi\int_X \frac{p_1(X)}{48}\right) . \tag{68}$$

The dependence on the argument of $m$ above means that the topological interpretation of the space of masses as a sphere is obstructed in the presence of a background metric.

#### 3.3.1   The Anomaly

We can, however, restore the identification of the points at infinite $|m|$ by introducing a suitable bulk term. Specifically we define a new partition function as

$$\tilde{Z}[m, g] = Z[m, g]\exp\left(2\pi i\int_Y \lambda(m) \wedge \frac{p_1(Y)}{48}\right) , \tag{69}$$

where above $\lambda(m)$ is any one-form which asymptotically approaches an angular form for large $|m|$:

$$\lim_{|m|\to\infty} \lambda(m) = \frac{d\arg(m)}{2\pi} \ . \tag{70}$$

The partition function $\tilde{Z}[m,g]$ is then invariant under phase rotation of $m$ for large $|m|$ and the topological interpretation of the spaces of masses as $\mathbb{S}^2$ is restored.

Observe that the anomaly (69) is supported by the non-trivial effective two-cycle of masses. In other words, if we restrict to any one-parameter slice of masses the anomaly trivializes. For instance along a circle of constant non-zero $|m|$ we can add to the Lagrangian a counterterm of the form change in the equation

$$\frac{i\arg(m)}{384\pi^2}\mathrm{Tr}(R\wedge R) \ , \tag{71}$$

and cancel the spurious transformation in (68). However, it is impossible to extend this counterterm to a smooth local $4d$ function of $m$ and $g$ on the entire two-dimensional $m$ plane. This obstruction is the anomaly.

As in the case of the $3d$ free fermion, we can also write the anomaly by applying the descent procedure a second time to obtain an anomaly six-form. In this case it is

$$\mathcal{I}_6 = \gamma(m) \wedge \frac{p_1(Y)}{48} \ , \tag{72}$$

where $\gamma = d\lambda$ is a two-form with total integral one on the mass-plane. (As above, in the free fermion theory it is natural to take $\gamma(m) = \delta^{(2)}(m)d^2m$, but below we will also discuss other natural forms.) This anomaly is similar to that found in the space of marginal coupling constants in [9, 10, 21]. See (203) for a determination of $\gamma(m)$ in a particular scheme and see section 7.4, in particular Remark 221, for further discussion.

The fact that $\gamma$ above has quantized total integral means that the anomaly is cohomologically non-trivial and hence it is preserved under continuous deformations of the theory including renormalization group flow. As with our discussion in previous sections, this also means that the same anomaly is present for more general interacting theories. For instance, analogously to (73) we can consider a theory with an additional real scalar $\varphi$

$$\mathcal{S} = \int d^4x \left( i\bar\psi\slashed\partial\psi + (\partial\varphi)^2 + [(m+\varphi)\psi\psi + c.c.] + V(\varphi) \right) \ . \tag{73}$$

This theory still has the anomaly (72) and hence the consequences discussed below.

### 3.3.2 Dynamical Consequences

We now apply the anomaly (72) to deduce general physical consequences. As always, we can consider a family of theories labelled by $m$ or a spacetime-dependent coupling $m(x)$.

- Non-trivial vacuum structure in codimension two: Consider the family of theories labelled by $m$ with an anomaly (72). Then, in order for the anomaly to be reproduced at long distances the theory cannot be trivially gapped for all $m$.

  Notice that unlike the discussion in sections 2 and 3.2, the non-trivial vacuum structure need only to occur in codimension two. In particular, this is the situation for the free fermion, which is everywhere trivially gapped except at the point $m = 0$. Thus, there is a non-trivial vacuum in the $m$-plane, but not necessarily a phase transition.

- Non-trivial strings: We can also consider space-dependent couplings where a two-cycle in spacetime wraps the $\mathbb{S}^2$ of mass parameters. For simplicity we consider a situation where the bulk is trivially gapped for generic $m$. In this case the anomaly (72) implies that there is a non-trivial effective string in the transverse space.[25] Specifically, by integrating the anomaly polynomial we find that wrapping $n$ times leads to an anomaly for the effective theory along the string

$$in \int_{Y_3} CS_{\text{grav}} \, . \tag{74}$$

Thus, the $2d$ theory on the string is gapless with chiral central charge $c_L - c_R = n/2$.

In the special case of the free fermion this conclusion can be readily verified by solving the Dirac equation in a background with position dependent mass as in [5,19,20], where one finds that the string supports $n$ Majorana-Weyl fermions in agreement with the general index theorems of [59,60].

As a simple special case of these general results, consider the mass profile

$$m(r,\theta) = \alpha r e^{i\theta} \, , \tag{75}$$

where $(r,\theta)$ parameterize a plane in radial coordinates and the string is localized along $r = 0$. We can split the $4d$ Weyl fermion into a left-moving $2d$ fermion $\psi_1$ and a right-moving $2d$ fermion $\psi_2$. Then one can check that in the mass profile (75) the field $\psi_2$ has no normalizable solutions and $\psi_1$ has only one normalizable solution

$$\psi_1 = c e^{-i\pi/4} e^{-\frac{1}{2}\alpha r^2} \, , \tag{76}$$

with a real coefficient $c$. Quantizing $c$ leads to one Majorana-Weyl fermion on the string worldvolume with chiral central charge $1/2$ as expected.

# 4 QED$_2$

In this section we explore the coupling anomalies in $2d$ $U(1)$ gauge theories. These models have a $\theta$-parameter and accordingly our analysis is similar to section 2. We refer to section 7.2 for additional discussion of the anomaly.

## 4.1 $2d$ Abelian Gauge Theory

We begin with $2d$ $U(1)$ gauge theory without charged matter. The Euclidean action is:

$$\mathcal{S} = \int \frac{1}{2g^2} da \wedge *da - \frac{i}{2\pi} \int \theta \, da \, . \tag{77}$$

Since the integral of $da$ is quantized, the transformation $\theta \to \theta + 2\pi$ does not affect the correlation functions of local operators at separated points. However, below we will show that the theories at $\theta$ and $\theta + 2\pi$ are only equivalent up to an invertible field theory.

The theory has a $U(1)$ one-form global symmetry associated to the two-form current $J \sim da$. This symmetry acts on the dynamical variable as $a \to a + \epsilon$ where $\epsilon$ is a flat connection. We can turn on a two-form background gauge field $B$ for this symmetry leading to the action

$$\mathcal{S} = \int \frac{1}{4g^2} (da - B) \wedge *(da - B) - \frac{i}{2\pi} \int \theta (da - B) - ik \int B \, , \tag{78}$$

---

[25]Following our discussion in the introduction, these are smooth external disturbances of the system, which are universal. These are not dynamical strings. If $m$ becomes a dynamical field, then, depending on the details of the theory, these strings could be stable dynamical objects.

where the coefficient $k$ of the counterterm is an integer. This action is invariant under background gauge transformation

$$a \to a + \Lambda, \quad B \to B + d\Lambda, \tag{79}$$

where $\Lambda$ is a $U(1)$ one-form gauge field. As in the comment following (35), we cannot use the classical $\Lambda$ to set the dynamical field $a$ to zero.

In the presence of nontrivial background gauge field $B$, the partition function $Z[\theta, B]$ is not invariant under $\theta \to \theta + 2\pi$. Instead, it satisfies

$$\frac{Z[\theta + 2\pi, B]}{Z[\theta, B]} = \exp\left(-i \int B\right). \tag{80}$$

This difference can be interpreted as an anomaly between the coupling $\theta$ and the $U(1)$ one-form global symmetry.

One can understand the anomaly more physically in terms of pair creation of probe particles, as in [43]. Adding to the action a $\theta$ term with coefficient $2\pi$ is equivalent to adding a Wilson line describing a pair of oppositely charged particles, which are created and then separated and moved to the boundary of spacetime. These particles screen the background electric field created by $\theta$, which is the physical reason for the $2\pi$ periodicity. However, when we take into account the one-form charge, the particle pair can be detected and this gives rise to the anomaly.

Extending the backgrounds $\theta$ and $B$ into a 3$d$ bulk $Y$ we can introduce a new partition function

$$\tilde{Z}[\theta, B] = Z[\theta, B] \exp\left(i \int \theta \, \frac{dB}{2\pi}\right), \tag{81}$$

which is invariant under $\theta \to \theta + 2\pi$.

### 4.1.1 Spacetime Dependent $\theta$

The anomaly can also be detected by promoting the coupling constant $\theta$ to be a variable function from spacetime to a circle. As in the discussion in section 2, our first task is to define more precisely the integral of $\theta \, da$ (and also the integral in (81)). This can be done precisely using the product in differential cohomology, as indicated in section 5.3.

Here, we can proceed as in section 2.1 and define the integral using patches. (This discussion seems more awkward than in section 2.1, but it is essentially the same as there.) Explicitly, we first cover spacetime by a collection of patches $\{U_I\}$. The circle-valued function $\theta$ can be lifted to real-valued functions on patches and transition functions between the patches:

$$\{\theta_I : U_I \to \mathbb{R}\} \quad \text{and} \quad \{n_{IJ} : U_I \cap U_J \to \mathbb{Z}\}, \qquad \text{with } \theta_I - \theta_J = 2\pi n_{IJ} \text{ on } U_I \cap U_J. \tag{82}$$

This data is redundant. If we modify

$$\theta_I \to \theta_I + 2\pi m_I, \qquad n_{IJ} \to n_{IJ} + m_I - m_J, \tag{83}$$

with integer $m_I$, we describe the same underlying circle-valued function $\theta$. Similarly the $U(1)$ gauge field $a$ can be lifted into the following data

$$\{a_I : U_I \to \Omega^1(U_I)\}, \quad \{\phi_{IJ} : U_I \cap U_J \to \mathbb{R}\} \quad \text{and} \quad \{n_{IJK} : U_I \cap U_J \cap U_J \to \mathbb{Z}\}, \tag{84}$$

where $\Omega^1(U_I)$ is the space of real differential one-forms on $U_I$. The lifts satisfy the following consistency conditions

$$\begin{aligned} U_I \cap U_J &: a_I - a_J = d\phi_{IJ}, \\ U_I \cap U_J \cap U_K &: \phi_{JK} + \phi_{KI} + \phi_{IJ} = 2\pi n_{IJK}, \end{aligned} \tag{85}$$

and there is a redundancy coming from gauge transformation

$$a_I \to a_I + d\lambda_I, \quad \phi_{IJ} \to \phi_{IJ} + \lambda_I - \lambda_J + 2\pi m_{IJ}, \quad n_{IJK} \to n_{IJK} + m_{JK} + m_{KI} + m_{IJ}, \quad (86)$$

where $\lambda_I$ are real functions on $U_I$ and $m_{IJ}$ are integers.

To define the integral, we need to pick a partition of spacetime into closed sets $\{\sigma_I\}$ with the properties: $\sigma_I \subset U_I$, $\sigma_{IJ} = (\partial\sigma_I \cap \partial\sigma_J) \subset U_I \cap U_J$ and $\sigma_{IJK} = (\partial\sigma_{IJ} \cap \partial\sigma_{JK} \cap \partial\sigma_{KI}) \subset U_I \cap U_J \cap U_K$. We define the integral of $\theta da$ in terms of the lifted data and the partition $\{\sigma_I\}$ as

$$\exp\left(\frac{i}{2\pi}\int \theta da\right) \equiv \exp\left(\frac{i}{2\pi}\sum_I \int_{\sigma_I} \theta_I da_I - i\sum_{I<J}\int_{\sigma_{IJ}} n_{IJ} a_J + i\sum_{I<J<K} n_{IJ}\phi_{JK}|_{\sigma_{IJK}}\right). \quad (87)$$

The first term in the right hand side is the naive expression. The second term is analogous to the similar term in (31) and the last term is needed to make the answer invariant under gauge transformations of $a$. One can check that this integral is independent of the choice of partitions $\{\sigma_I\}$ and the lifts of $\theta$ and $A$.

Similarly, the integral (81) should be defined more carefully when $\theta$ varies in spacetime.

In a configuration where $\theta$ has non-trivial winding number along some one-cycle the resulting integral breaks the one-form global symmetry. As an illustration, consider a simple situation where spacetime is a torus with one-cycles $x$ and $y$ and $\theta$ has winding number $m$ around $x$ and is independent of $y$. If we restrict to a sector with $\int_{T^2} da = 0$ then we have

$$\exp\left(\frac{i}{2\pi}\int_{T^2}\theta da\right) = \exp\left(im\oint_y a\right). \quad (88)$$

The Wilson line on the right-hand side above is charged under the one-form symmetry (79) thus illustrating the breaking. One way to think about this breaking is to note that for this configuration of spacetime dependent $\theta$ nonzero correlation functions must involve an appropriate net number of Wilson lines circling the $y$-cycle.

As in our previous discussion, we can restore the invariance under the one-form symmetry by coupling to a bulk using the partition function $\tilde{Z}$ in (81). For instance, in the torus example above we can extend the background fields to a three-manifold $Y$ which is a solid torus with the cycle $y$ filled in to a disk $D$. We then evaluate the anomaly[26]

$$\exp\left(i\int_Y \theta\frac{dB}{2\pi}\right) = \exp\left(-im\int_D B\right). \quad (89)$$

The combination of (88) and (89) is then invariant under the one-form gauge transformations (79).

We can also express this violation of the one-form symmetry and the anomaly (80) in terms of a four-form using the descent procedure

$$\mathcal{I}_4 = \frac{1}{(2\pi)^2}d\theta \wedge dB. \quad (90)$$

This is the curvature of the anomaly theory identified in (192). We remark that as discussed above, $d\theta$ can be replaced by $d(\theta + f(\theta))$ with an arbitrary $2\pi$-periodic function $f(\theta)$.

---

[26]The equation (89) is correct up to a boundary term $\exp\left(\frac{i}{2\pi}\int_{T^2}\theta B\right)$ which cancels against a similar term in the action (78).

### 4.1.2 Dynamics

The 2d $U(1)$ gauge theory has no local degrees of freedom – it is locally trivial. In the spirit of the 't Hooft anomaly matching the non-trivial anomaly in (81) or equivalently (90) must be reproduced by its effective description. As a result, the theory cannot be completely trivial for all values of $\theta$. Indeed, as we will now review, it has a first order phase transition at $\theta = \pi$.

We can say more about the dynamics using charge conjugation $\mathcal{C}$, which is a symmetry when $\theta = 0$ or $\theta = \pi$. This symmetry acts as

$$\mathcal{C}(a) = -a, \quad \mathcal{C}(B) = -B. \tag{91}$$

At $\theta = \pi$, the charge conjugation symmetry is accompanied by a $2\pi$-shift of $\theta$ and this leads to a mixed anomaly between $\mathcal{C}$ and the one-form symmetry [23,36]. Indeed, using (80) we see that when $\theta = \pi$, a $\mathcal{C}$ transformation acts on the partition function as

$$Z[\pi, B] \to Z[\pi, -B] = Z[-\pi, -B] \exp\left(i \int B\right) = Z[\pi, B] \exp\left((1 - 2k)i \int B\right), \tag{92}$$

and we cannot choose $k$ to remove this transformation since $k$ is required to be integral. This obstruction characterizes the $\mathcal{C}$ anomaly. As above, this anomaly can be written using inflow as

$$\mathcal{A}(C, B) = \exp\left(\frac{i}{2}\int_Y dB + C \cup B\right), \tag{93}$$

where $C$ is a $\mathbb{Z}_2$ gauge field for charge conjugation (with holonomies 0, $\pi$).[27]

The anomaly involving $\mathcal{C}$ at $\theta = \pi$ implies that the long-distance theory for this value of $\theta$ cannot be trivially gapped. This agrees with the fact that the charge conjugation symmetry $\mathcal{C}$ at $\theta = \pi$ is spontaneously broken. The $U(1)$ one-form symmetry cannot be spontaneously broken in 2d and the theory is gapped at long distance. Thus, the anomaly can only be saturated by the spontaneously broken charge conjugation symmetry. The anomalies and their consequences are summarized in Table 2.

Of course, this system is exactly solvable and this analysis of its symmetries and anomalies does not lead to any new results. However, as we will soon see, the same reasoning leads to new results in more complicated systems, which are not exactly solvable.

The second class of consequences is associated with defects where $\theta$ varies in space. Let us first place the theory on $\mathbb{S}^1 \times \mathbb{R}$ with a constant $\theta$. The effective quantum mechanics is the particle on a circle studied in section 2 with $q = \oint A$ the holonomy of $A$ along the circle. The anomaly involving $\theta$ discussed above reduces to the anomaly (39) between $\theta$ and the $U(1)$ global symmetry in the quantum mechanics.

Next, we also let $\theta$ vary along the $\mathbb{S}^1$ direction and insert Wilson lines $\exp(i \int k_I A(x_I))$ along the $\mathbb{R}$ direction. In Lorentzian signature, the path integral over $A_t$ imposes the Gauss constraint

$$\partial_x F_{xt} = g^2 \left(\sum k_I \delta(x - x_I) - \frac{\partial_x \theta(x)}{2\pi}\right), \tag{94}$$

and therefore $\partial_x \theta$ can be interpreted as a space-dependent background charge density. Integrating the constraint we learn that the total background charge density has to vanish

$$2\pi \sum k_I - \int \partial_x \theta = 0. \tag{95}$$

This implies that the theory is not consistent on a compact space where $\theta$ has a nontrivial winding unless there are Wilson lines inserted to absorb the charge.

---

[27]If $C \to Y$ is the double cover defining the $\mathbb{Z}_2$ gauge field, then because of the twisting (91) the characteristic class of $B$ lies in twisted cohomology: $[B] \in H^3(Y; \mathbb{Z}_C)$. The partition function is the value of the mod 2 reduction $\overline{[B]} \in H^3(Y; \mathbb{Z}_2)$ on the fundamental class of $Y$; see (194).

Table 2: Summary of anomalies and existence of continuous counterterms preserving $\mathcal{C}$ ("global inconsistency") in various $2d$ theories. The superscripts of the symmetries label the $q$'s of $q$-form symmetries. All these theories have a charge conjugation symmetry, $\mathcal{C}$, at $\theta = 0, \pi$. Without using the charge conjugation symmetry, all these theories exhibit a mixed anomaly involving the coupling $\theta$ and some global symmetry $\mathcal{G}$. The anomaly implies that the long distance theory cannot be trivially gapped everywhere between $\theta$ and $\theta + 2\pi$. By including $\mathcal{C}$ we see that the theories can have a mixed anomaly between $\mathcal{C}$ and some global symmetry $\mathcal{G}$ at $\theta = \pi$. Such an anomaly forbids the long distance theories to be trivially gapped at $\theta = \pi$. Even if the theories have no mixed anomaly at $\theta = \pi$, there may be no smooth counterterms that preserve $\mathcal{C}$ simultaneously at $\theta = 0$ and $\theta = \pi$. Finally, we can deform these systems and break $\mathcal{C}$. Then the results in the "without $\mathcal{C}$" column are still applicable. The only difference is that we do not know at what value of $\theta$ the transition takes place.

| theory | without $\mathcal{C}$ | with $\mathcal{C}$ at $\theta = 0, \pi$ | |
|---|---|---|---|
| symmetry $\mathcal{G}$ | $\theta$-$\mathcal{G}$ anomaly | $\mathcal{C}$-$\mathcal{G}$ anomaly at $\theta = \pi$ | no smooth counterterm |
| $U(1)$ gauge theory $\mathcal{G} = U(1)^{(1)}$ | ✓ | ✓ | ✓ |
| with 1 charge $p$ scalar $\mathcal{G} = \mathbb{Z}_p^{(1)}$ , $p > 1$ | ✓ | even $p$ ✓  odd $p$ ✗ | even $p$ ✓  odd $p$ ✓ |
| with $N$ charge 1 scalar $\mathcal{G} = PSU(N)^{(0)}$ , $N > 1$ | ✓ | even $N$ ✓  odd $N$ ✗ | even $N$ ✓  odd $N$ ✓ |

## 4.2 QED$_2$ with one charge $p$ scalar

We now add to the theory a scalar of charge $p$. (See [61–64] for early discussion of this theory.) The Euclidean action becomes

$$S = \int \frac{1}{2g^2} da \wedge *da - \frac{i}{2\pi} \int \theta \, da + \int d^2x \left( |D_{pa}\phi|^2 + V(|\phi|^2) \right) . \tag{96}$$

The charge $p$ scalar breaks the $U(1)$ one-form symmetry to a $\mathbb{Z}_p$ one-form symmetry [1]. As before, we can activate the background gauge field $K \in H^2(X, \mathbb{Z}_p)$ for this symmetry and the modified action includes

$$S \supset -\frac{i}{2\pi} \int \theta \left( da - \frac{2\pi}{p} K \right) - \frac{2\pi i k}{p} \int K , \tag{97}$$

where the coefficient of the counterterm $k$ is an integer modulo $p$.

The theory has a mixed anomaly between the coupling $\theta$ and the $\mathbb{Z}_p$ one-form symmetry. The $3d$ anomaly is

$$\mathcal{A}(\theta, K) = \exp\left( -2\pi i \int \frac{d\theta}{2\pi} \frac{K}{p} \right) , \tag{98}$$

(see the discussion below (44) for comments on boundary terms.) The anomaly forces the long distance theory to be nontrivial for at least one-point between $\theta$ and $\theta + 2\pi$.

The same conclusion can be drawn from Hamiltonian formalism. We can decompose the Hilbert space of the theory into superselection sectors according to the $\mathbb{Z}_p$ one-form symmetry [64]

$$\mathcal{H} = \bigoplus_{n=1}^{p} \mathcal{H}_n . \tag{99}$$

Intuitively, transitions using Coleman's pair-creation mechanism [43] using the dynamical quanta can change $\theta$ by $2\pi p$ and hence they take place within one of the subspaces in (99). But transitions between states in different subspaces labeled by different values of $n$ in (99) can take place only using probe particles. As a result, all the subspaces in (99) are in the same theory but time evolution preserves the subspace [64].

The Hilbert spaces at $\theta$ and $\theta + 2\pi$ are isomorphic but the superselection sectors are shuffled. In particular this means that the vacuum at $\theta$ is no longer the vacuum at $\theta + 2\pi$ and therefore the long distance theory cannot be trivial everywhere between $\theta$ and $\theta + 2\pi$.

For smooth interfaces interpolating between $\theta$ and $\theta + 2\pi n$, the bulk anomaly (98) yields the effective anomaly of the interface

$$\mathcal{A}(K) = \exp\left(-2\pi i n \int \frac{K}{p}\right). \tag{100}$$

### 4.2.1 Implications of $\mathcal{C}$ Symmetry

The discussion above did not make use of the charge conjugation symmetry $\mathcal{C}$ at $\theta = 0, \pi$, and as usual we can say more using this additional symmetry. $\mathcal{C}$ acts as

$$\mathcal{C}(a) = -a, \quad \mathcal{C}(\phi) = \phi^*, \quad \mathcal{C}(K) = -K, \tag{101}$$

and at $\theta = \pi$, the partition function transforms as

$$Z[\pi, K] \to Z[\pi, K] \exp\left(2\pi i \frac{1-2k}{p} \int K\right). \tag{102}$$

Since the coefficient of the counterterm is an integer modulo $p$, the partition function transforms non-anomalously if there is an integer $k$ that solves

$$2k = 1 \bmod p. \tag{103}$$

For even $p$, there are no solutions and the charge conjugation symmetry has a mixed anomaly with the $\mathbb{Z}_p$ one-form symmetry at $\theta = \pi$.[28] The anomaly enforces non-trivial long distance physics at $\theta = \pi$.

For odd $p$, the condition (103) can be solved by $k = \frac{p+1}{2}$ so there is no anomaly at $\theta = \pi$. We can however make a weaker statement by noticing that the counterterm that preserves charge conjugation symmetry at $\theta = 0$, has coefficient $k = 0$ and it differs from the choice of counterterm at $\theta = \pi$. Similar phenomena have been discussed in [23, 36, 38]. In [38, 40], this situation was referred to as a "global inconsistency." Concretely it means that there is no continuously varying ($\theta$-dependent) counterterm that preserves $\mathcal{C}$ at both $\theta = 0$ and $\theta = \pi$. This again implies that the long-distance theory is non-trivial for at least one value of $\theta$ in $[0, \pi]$. This discussion is summarized in Table 2.

All these constraints are saturated by spontaneously broken charge conjugation symmetry at $\theta = \pi$. The special value $p = 1$ deserves further comment. In this case, there is no one-form symmetry so the constraints described above no longer hold. If the scalar is very massive, the theory is effectively a pure $U(1)$ gauge theory so the theory is gapped for generic $\theta$ and the charge conjugation symmetry is spontaneously broken at $\theta = \pi$ leading to a first order phase transition. On the other hand, if the scalar condenses, the gauge field is Higgsed and the theory is trivially gapped for all $\theta$.[29] Therefore, the line of first order phase transitions at $\theta = \pi$ must end at some intermediate value of the mass where the theory is gapless.

---

[28] The anomaly is $\mathcal{A}(C, K) = \exp\left(i\pi \int C \cup K\right)$ where $C$ is the $\mathbb{Z}_2$ charge conjugation gauge field. Note that this is meaningful only when $p$ is even.

[29] In the limit of large scalar expectation value the smooth $\theta$-dependence of various observables is reliably computed using instanton methods. These techniques are not reliable in the opposite limit of large mass for the scalar. And indeed, in that limit the $\theta$-dependence is not smooth.

### 4.3 QED$_2$ with $N$ charge 1 scalars

We now add $N$ charge 1 scalars into the $U(1)$ gauge theory. The Euclidean action is

$$S = \int \frac{1}{2g^2} da \wedge *da - \frac{i}{2\pi} \int \theta \, da + \int d^2 x \left[ \sum_{I=1}^{N} |D_a \phi_I|^2 + V \left( \sum_{I=1}^{N} |\phi_I|^2 \right) \right]. \qquad (104)$$

If the potential $V(\sum |\phi_I|^2)$ has a minimum at $\sum |\phi_I|^2 \neq 0$ and is sufficiently steep, the above theory flows to a $\mathbb{CP}^{N-1} = \frac{U(N)}{U(N-1) \times U(1)}$ non-linear sigma model.[30]

The $U(1)$ one-form symmetry is now completely broken. Instead the theory has a $PSU(N) \cong SU(N)/\mathbb{Z}_N$ zero-form global symmetry that acts as $\phi_I \to \mathcal{G}_{IJ} \phi_J$. The reason the symmetry is $PSU(N)$ and not simply $SU(N)$ is that the $\mathbb{Z}_N$ transformation $\phi_I \to e^{2\pi i/N} \phi_I$ coincides with a $U(1)$ gauge transformation and hence acts trivially on all gauge invariant local operators.

Let us consider the system in the presence of a background gauge field $A$ for the $PSU(N)$ global symmetry. The correlation of center of $SU(N)$ with the dynamical $U(1)$ gauge group means that $a$ and $A$ combine to a connection for the group

$$U(N) = \frac{SU(N) \times U(1)}{\mathbb{Z}_N}. \qquad (105)$$

Crucially this means that in a general $PSU(N)$ background, $a$ is no longer a $U(1)$ connection with properly quantized fluxes. Instead we have

$$\oint \frac{da}{2\pi} = \oint \frac{w_2(A)}{N} \bmod 1, \qquad (106)$$

where $w_2(A) \in H^2(X, \mathbb{Z}_N)$ is the second Stiefel-Whitney class of the $PSU(N)$ bundle. Equivalently, in the presence of general $PSU(N)$ backgrounds there are fractional instantons.

Because of these fractional instantons, the partition function is no longer invariant under $\theta \to \theta + 2\pi$. Rather, $\theta$ has an extended periodicity of $2\pi N$ [23, 36, 37]. This represents a mixed anomaly between the $2\pi$-periodicity of $\theta$ and the $PSU(N)$ global symmetry. The corresponding $3d$ anomaly is

$$\mathcal{A}(\theta, A) = \exp \left( 2\pi i \int \frac{d\theta}{2\pi} \frac{w_2}{N} \right), \qquad (107)$$

(see the discussion below (44) for a comment on the boundary terms). The anomaly implies that the long distance theory cannot be trivially gapped everywhere between $\theta$ and $\theta + 2\pi$.

Like the discussion in sections 4.1.2 and 4.2, we can understand this anomaly physically in terms of particle pair creation following [43]. The $\theta$-term with coefficient $2\pi$ can be screened to $\theta = 0$ by pair creation of dynamical quanta. (Note that in the discussion in section 4.1.2 we used probe particles, and in section 4.2 we discussed the effects of both dynamical and probe quanta.) These quanta transform projectively under $PSU(N)$ and hence the screening leads to such projective representations at the boundary of space. More mathematically, this is the meaning of the selection rule (106).

It is interesting to compare this discussion with the anomaly between $\theta$-periodicity and the one-form global symmetry in section 4.2. The role of the background two-form $\mathbb{Z}_p$ gauge field $K$ there is played here by the background $w_2$ associated with the zero-form $PSU(N)$ global symmetry.

---

[30]We can easily generalize our analysis below to systems with several $U(1)$ gauge fields and various charged scalars. In that case the systems have several $\theta$-parameters. Recently studied examples include systems that flow to $2d$ sigma-models whose target space is the flag manifold $\frac{U(N)}{U(N_1) \times \cdots \times U(N_m)}$ [39, 65–67].

### 4.3.1 Implications of $\mathcal{C}$ Symmetry

We can further constrain the long distance theory using the charge conjugation symmetries $\mathcal{C}$ at $\theta = 0, \pi$, which acts as

$$\mathcal{C}(a) = -a, \quad \mathcal{C}(\phi_I) = \phi_I^*, \quad \mathcal{C}(A) = -A, \quad \mathcal{C}(w_2(A)) = -w_2(A). \tag{108}$$

We can add to the theory a counterterm

$$\mathcal{S} \supset -2\pi i \frac{k}{N} \int w_2. \tag{109}$$

At $\theta = \pi$, the charge conjugation symmetry involves a $2\pi$-shift of $\theta$ and it transforms the partition function as

$$Z[\pi, A] \to Z[\pi, A] \exp\left(2\pi i \frac{1 - 2k}{N} \int w_2\right). \tag{110}$$

Similar to the example of $QED_2$ with one charge $p$ scalar discussed in the last subsection, the above means that charge conjugation symmetry has a mixed anomaly with the $PSU(N)$ global symmetry for even $N$.[31] Meanwhile for odd $N$, there is no continuous counterterm preserving $\mathcal{C}$ at both $\theta = 0$ and $\theta = \pi$. For even $N$, the anomaly forces a non-trivial long distance theory at $\theta = \pi$, while for odd $N$ we find a non-trivial theory for at least one value of $\theta$.

These constraints agree with the common lore. For $N \geqslant 2$, the theory is believed to be gapped at generic $\theta$ except at $\theta = \pi$. For $N = 2$, the model at $\theta = \pi$ flows to the $SU(2)_1$ WZW model [68]. For $N > 2$, the charge conjugation symmetry is believed to be spontaneously broken at $\theta = \pi$ [69]. (The model with $N = 1$ was discussed above.)

Finally, we can consider a smooth interface between $\theta$ and $\theta + 2\pi n$. Assuming the theory is gapped for generic $\theta$, at long distances there is then an isolated quantum mechanics on the interface. The anomaly (107) implies that the quantum mechanical model has a non-trivial anomaly for the $PSU(N)$ global symmetry encoded by

$$\mathcal{A}(A) = \exp\left(2\pi i \int n \frac{w_2}{N}\right). \tag{111}$$

This means that the ground states of the quantum mechanics are degenerate, and they form a projective representation of the $PSU(N)$ symmetry (i.e. a representation of $SU(N)$) with $N$-ality $n$. Intuitively, the interface is associated with $n\, \Phi_I$ quanta. But since they are strongly interacting we cannot determine their precise state except their $N$-ality.

## 5 Introduction to Differential Cohomology

Generalized differential cohomology is the formalism we use to express invertible field theories, so in this section we provide an expository introduction to the subject. For another expository introduction, see [70].

Ordinary differential cohomology groups were introduced under the name *differential characters* by Cheeger-Simons [71] in the early 1970s. Differential cohomology groups may be viewed as an analog of Deligne cohomology [72] for smooth manifolds; see also [73, §6.3]. The work of Hopkins-Singer [46] extends the differential theory to generalized cohomology theories and also develops a version with geometric representatives of differential cohomology classes. See [47–49] and the references therein for modern developments. Our discursive

---

[31]The anomaly is $\exp\left(i\pi \int C \cup w_2\right)$ where $C$ is the charge conjugation gauge field. This is meaningful only when $N$ is even.

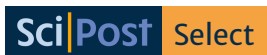

treatment of low degree classes is meant as background for the reader, who should pursue the subject in these and other references.

Ordinary differential cohomology attaches a sequence of (infinite dimensional) abelian Lie groups $\mathcal{A}^k(M)$, $k = 0, 1, 2, \ldots$, to each smooth manifold $M$. The group $\mathcal{A}^0(M) = H^0(M; \mathbb{Z})$ is the group of integer-valued smooth—so necessarily locally constant—functions $M \to \mathbb{Z}$. In degrees $k = 1$ and $k = 2$ we can also give concrete descriptions of $\mathcal{A}^k(M)$, to which we now turn.

## 5.1 Hermitian Line Bundles with Covariant Derivative

Fix a smooth manifold $M$ and let $\mathcal{A}^2(M)$ denote the set of isomorphism classes of hermitian line bundles $\pi \colon L \to M$ equipped with compatible covariant derivative $\nabla$. Then $\mathcal{A}^2(M)$ is an abelian group under tensor product. We break $\mathcal{A}^2(M)$ down in several ways. First, the curvature of $\nabla$ is an imaginary 2-form which adds under tensor product, so curvature times[32] $\sqrt{-1}/2\pi$ is a homomorphism

$$\omega \colon \mathcal{A}^2(M) \longrightarrow \Omega^2_{\text{closed}}(M). \tag{112}$$

The kernel of $\omega$ is the subgroup $\mathcal{A}^2_{\text{flat}}(M) \subset \mathcal{A}^2(M)$ of isomorphism classes of flat hermitian line bundles. The de Rham cohomology class of $\sqrt{-1}/2\pi$ times the curvature is the (real) Chern class of $\pi \colon L \to M$, so the image of the map (112) is the union of affine translates of the subspace of exact 2-forms $d\Omega^1(M) \subset \Omega^2_{\text{closed}}(M)$. These affine spaces are parametrized by their de Rham cohomology classes, which form a full lattice in the second de Rham cohomology. The lattice is the image of

$$H^2(M; \mathbb{Z}) \longrightarrow H^2(M; \mathbb{R}). \tag{113}$$

The kernel of (113) is the subgroup $\operatorname{Tors} H^2(M; \mathbb{Z}) \subset H^2(M; \mathbb{Z})$ of torsion elements. The Chern class homomorphism $c$ forgets the covariant derivative; its compatibility with (112) is expressed by the commutative diagram

$$
\begin{array}{ccc}
\mathcal{A}^2(M) & \overset{\omega}{\longrightarrow} & \Omega^2_{M,\text{closed}} \\
{\scriptstyle c}\big\downarrow & & \big\downarrow \\
H^2(M; \mathbb{Z}) & \longrightarrow & H^2(M; \mathbb{R})
\end{array}
\tag{114}
$$

The group of flat covariant derivatives is

$$\mathcal{A}^2_{\text{flat}}(M) \cong H^1(M; \mathbb{R}/\mathbb{Z}) \cong \operatorname{Hom}(H_1 M, \mathbb{R}/\mathbb{Z}), \tag{115}$$

where the isomorphism maps a flat covariant derivative to its holonomy, a function on loops in $M$. The Chern class of a flat hermitian line bundle lies in the torsion subgroup of $H^2(M; \mathbb{Z})$, and the group of flat covariant derivatives on the trivial line bundle is the torus

$$T^1(M) = \frac{H^1(M; \mathbb{R})}{H^1(M; \mathbb{Z})}. \tag{116}$$

The situation is summarized by the short exact sequence

$$0 \longrightarrow T^1(M) \longrightarrow \mathcal{A}^2_{\text{flat}}(M) \longrightarrow \operatorname{Tors} H^2(M; \mathbb{Z}) \longrightarrow 0. \tag{117}$$

---

[32]To handle the factors of $2\pi\sqrt{-1}$ more gracefully, we do the Tate twist in section 5.5 below.

**Example 118.** For the circle $M = S^1$ the group $\mathcal{A}^2(S^1) = T^1(S^1)$ is isomorphic to $\mathbb{R}/\mathbb{Z}$; the isomorphism maps a bundle with covariant derivative to its holonomy. The Chern class is necessarily zero. On the other hand, for the real projective plane $M = \mathbb{RP}^2$ the torus $T^1(\mathbb{RP}^2)$ is trivial and $\mathcal{A}^2(\mathbb{RP}^2) \cong \operatorname{Tors} H^2(\mathbb{RP}^2; \mathbb{Z})$ is isomorphic to $\mathbb{Z}/2\mathbb{Z}$. In this case the nontrivial element is detected by the Chern class. The topologically nontrivial hermitian line bundle over $\mathbb{RP}^2$ admits a unique flat covariant derivative, up to gauge transformations, as does the trivial hermitian line bundle.

These decompositions tell the structure of $\mathcal{A}^2(M)$ as an abelian group. Furthermore, $\mathcal{A}^2(M)$ can be given the structure of an infinite dimensional *abelian Lie group*. Its Lie algebra is

$$\operatorname{Lie} \mathcal{A}^2(M) \cong \frac{\Omega^1(M)}{d\Omega^0(M)}, \tag{119}$$

with trivial bracket, and its nonzero homotopy groups are

$$\pi_0 \mathcal{A}^2(M) \cong H^2(M; \mathbb{Z}), \tag{120}$$

$$\pi_1 \mathcal{A}^2(M) \cong H^1(M; \mathbb{Z}). \tag{121}$$

Each component of $\mathcal{A}^2(M)$ is a principal $T^1(M)$-bundle over an affine translate of $d\Omega^1(M)$.

*Remark* 122. Notice the distinction between *isomorphism classes* and *deformation classes*. The abelian group of isomorphism classes of hermitian line bundles with covariant derivative is $\mathcal{A}^2(M)$, whereas the abelian group of deformation classes is the group $H^2(M; \mathbb{Z})$ of path components of $\mathcal{A}^2(M)$. The latter does not track the local differential-geometric data—the covariant derivative and curvature—whereas the former does. In Example 118 every element of $\mathcal{A}^2(S^1)$ is deformation equivalent to zero, even if the isomorphism class is nonzero. On the other hand, the group $\mathcal{A}^2(\mathbb{RP}^2)$ is discrete and so isomorphism classes and deformation classes coincide. Said differently, $\mathcal{A}^2(\mathbb{RP}^2) = \mathcal{A}^2_{\mathrm{flat}}(\mathbb{RP}^2)$. We see that in general the deformation class of $(\pi \colon L \to M, \nabla)$ retains only the topological information of the line bundle $\pi \colon L \to M$, whereas the isomorphism class remembers geometric information encapsulated in the covariant derivative $\nabla$ as well.

## 5.2 Trivializations

An important observation: we cannot define trivializations of isomorphism classes. After all, a trivialization is an isomorphism with the trivial bundle with trivial covariant derivative, but there is no notion of a map between isomorphism classes. Hence to define trivializations we work directly with geometric objects.[33] Then there are two sorts of trivializations in differential cohomology.

**Definition 123.** Let $M$ be a smooth manifold, $\pi \colon L \to M$ a hermitian line bundle, and $\nabla$ a compatible covariant derivative. Denote this data as $(\pi, \nabla)$.

(i) A *flat trivialization* of $(\pi, \nabla)$ is a section $\tau \colon M \to L$ of $\pi$ such that $|\tau| = 1$ and $\nabla \tau = 0$,

(ii) A *nonflat trivialization* of $(\pi, \nabla)$ is a section $\tau \colon M \to L$ of $\pi$ such that $|\tau| = 1$.

There is no constraint on the covariant derivative of a nonflat trivialization. The norm condition is innocuous—we can normalize any nonzero section. Standard elementary arguments determine the obstructions to the existence of these trivializations. Let $[\pi, \nabla] \in \mathcal{A}^2(M)$ denote the isomorphism class of $(\pi, \nabla)$, and let $[\pi] \in H^2(M; \mathbb{Z})$ denote its deformation class, which is the Chern class of $\pi \colon L \to M$. Then

---

[33]The collection of these objects forms a *category*—in this case a *Picard groupoid*—which includes the data of maps between objects and a distinguished trivial (unit) object.

(i) a flat trivialization exists if and only if $[\pi, \nabla] = 0$ in $\mathcal{A}^2(M)$,

(ii) a nonflat trivialization exists if and only if $[\pi] = 0$ in $H^2(M; \mathbb{Z})$.

Each species of trivialization with its corresponding obstruction has an echo in the context of invertible field theories, and so pertains to the study of anomalies.

Suppose $\tau$ is a nonflat trivialization of $(\pi, \nabla)$. Define the global 1-form $\alpha \in \Omega^1(M)$ by

$$\frac{\sqrt{-1}}{2\pi} \nabla \tau = \alpha \tau. \tag{124}$$

Up to isomorphism the triple $(\pi, \nabla, \tau)$ is equivalent to the 1-form $\alpha$.

Before considering ratios of trivializations, we introduce

$$\mathcal{A}^1(M) = \mathrm{Map}(M, \mathbb{R}/\mathbb{Z}), \tag{125}$$

the abelian Lie group of smooth functions from $M$ into the circle. It is the first differential cohomology group of $M$. In degree 1, as in degree 0, there is no equivalence relation on the geometric objects, which for $\mathcal{A}^1(M)$ are smooth functions $M \to \mathbb{R}/\mathbb{Z}$. The structure of $\mathcal{A}^1(M)$ is similar to that of $\mathcal{A}^2(M)$, but with downshifted degrees. For example, analogous to (119)–(121) we have

$$\mathrm{Lie}\, \mathcal{A}^1(M) \cong \Omega^0(M), \tag{126}$$

$$\pi_0 \mathcal{A}^1(M) \cong H^1(M; \mathbb{Z}), \tag{127}$$

$$\pi_1 \mathcal{A}^1(M) \cong H^0(M; \mathbb{Z}), \tag{128}$$

and analogous to (115) the flat elements

$$\mathcal{A}^1_{\mathrm{flat}}(M) \cong H^0(M; \mathbb{R}/\mathbb{Z}) \tag{129}$$

form the abelian group of locally constant circle-valued functions.[34]

Returning to trivializations of $(\pi\colon L \to M, \nabla)$ we see that

(i) the ratio of two flat trivializations is an element of $\mathcal{A}^1_{\mathrm{flat}}(M)$,

(ii) the ratio of two nonflat trivializations is an element of $\mathcal{A}^1(M)$.

Equivalently,

(i) the space of flat trivializations is a *torsor* over $\mathcal{A}^1_{\mathrm{flat}}(M)$,

(ii) the space of nonflat trivializations is a torsor over $\mathcal{A}^1(M)$.

## 5.3 Multiplication

There is a multiplication map

$$\cdot \colon \mathcal{A}^{k_1}(M) \times \mathcal{A}^{k_2}(M) \longrightarrow \mathcal{A}^{k_1 + k_2}(M) \tag{131}$$

on differential cohomology for any nonnegative integers $k_1, k_2$. The first nontrivial case

$$\cdot \colon \mathcal{A}^1(M) \times \mathcal{A}^1(M) \longrightarrow \mathcal{A}^2(M) \tag{132}$$

---

[34]Special to this degree is the fact that $H^1(M; \mathbb{Z})$ is torsionfree, so $\mathcal{A}^1_{\mathrm{flat}}(M)$ is the connected torus

$$T^0(M) = \frac{H^0(M; \mathbb{R})}{H^0(M; \mathbb{Z})}. \tag{130}$$

Put differently, every locally constant circle-valued function has a lift to a real-valued function. By contrast, there exist manifolds $M$ and topologically nontrivial hermitian line bundles $\pi\colon L \to M$ which admit a flat covariant derivative, as in Example 118.

has an explicit geometric model as follows. An element of $\mathcal{A}^1(M)$ is a map $M \to \mathbb{R}/\mathbb{Z}$, so the universal version of (132) is the product in $\mathcal{A}^\bullet(T)$ of the projection maps onto the factors of the torus $T = \mathbb{R}/\mathbb{Z} \times \mathbb{R}/\mathbb{Z}$. That product is the isomorphism class of a hermitian line bundle with covariant derivative $(\pi_T, \nabla_T)$ on $T$, which we now specify. Let $\phi^1, \phi^2$ be standard coordinates on $\mathbb{R} \times \mathbb{R}$ and $\overline{\phi}^1, \overline{\phi}^2$ their reductions to $\mathbb{R}/\mathbb{Z}$, which pass to functions $T \to \mathbb{R}/\mathbb{Z}$. The pair $(\pi_T, \nabla_T)$ is characterized up to isomorphism by:

- the curvature is $d\overline{\phi}^1 \wedge d\overline{\phi}^2$

- the holonomy around the circle $\mathbb{R}/\mathbb{Z} \times \{0\}$ equals one

- the holonomy around the circle $\{0\} \times \mathbb{R}/\mathbb{Z}$ equals one

$$(133)$$

The general product in (132) is obtained by pullback: if $\overline{\phi}^1, \overline{\phi}^2 : M \to \mathbb{R}/\mathbb{Z}$, and $[\overline{\phi}^1], [\overline{\phi}^2] \in \mathcal{A}^1(M)$ the corresponding degree one differential cohomology classes, then the product

$$[\overline{\phi}^1] \cdot [\overline{\phi}^2] = (\overline{\phi}^1 \times \overline{\phi}^2)^* [\pi_T, \nabla_T] \tag{134}$$

is the pullback of the isomorphism class of $(\pi_T, \nabla_T)$ by the map $\overline{\phi}^1 \times \overline{\phi}^2 : M \to T$.

We elucidate several features of this construction which have analogs for the product (131) in any degree. First, the Chern class of $\pi_T$ is the cup product of the generators of the cohomology groups $H^1(\mathbb{R}/\mathbb{Z} \times \{0\}; \mathbb{Z})$ and $H^1(\{0\} \times \mathbb{R}/\mathbb{Z}; \mathbb{Z})$ of the "axes" of the torus $T$. It follows that the product (132) is compatible with cup product via the Chern class maps, i.e., the diagram

$$
\begin{array}{ccc}
\mathcal{A}^1(M) \times \mathcal{A}^1(M) & \xrightarrow{\ \cdot\ } & \mathcal{A}^2(M) \\
{\scriptstyle c \times c} \downarrow & & \downarrow {\scriptstyle c} \\
H^1(M;\mathbb{Z}) \times H^1(M;\mathbb{Z}) & \xrightarrow{\ \smile\ } & H^2(M;\mathbb{Z})
\end{array}
\tag{135}
$$

commutes. Similarly, the curvature $d\overline{\phi}^1 \wedge d\overline{\phi}^2$ of $\nabla_T$ is the product of the standard 1-forms on the axes, which implies that the product (132) is compatible with wedge product: the diagram

$$
\begin{array}{ccc}
\mathcal{A}^1(M) \times \mathcal{A}^1(M) & \xrightarrow{\ \cdot\ } & \mathcal{A}^2(M) \\
{\scriptstyle \omega \times \omega} \downarrow & & \downarrow {\scriptstyle \omega} \\
\Omega^1_{\text{closed}}(M) \times \Omega^1_{\text{closed}}(M) & \xrightarrow{\ \wedge\ } & \Omega^2_{\text{closed}}(M)
\end{array}
\tag{136}
$$

commutes.

Next, suppose one of the circle-valued maps in (134), say $\overline{\phi}^1 : M \to \mathbb{R}/\mathbb{Z}$, is lifted to a real-valued function $\phi^1 : M \to \mathbb{R}$. Then $\overline{\phi}^1 \times \overline{\phi}^2 : M \to T = \mathbb{R}/\mathbb{Z} \times \mathbb{R}/\mathbb{Z}$ lifts to $\phi^1 \times \overline{\phi}^2 : M \to \widetilde{T} = \mathbb{R} \times \mathbb{R}/\mathbb{Z}$. On the cylinder $\widetilde{T}$ the pullback curvature form $d\phi^1 \wedge d\overline{\phi}^2$ has a global antiderivative, namely the 1-form $\phi^1 d\overline{\phi}^2$. Furthermore, any other antiderivative, modulo the group $d\Omega^0(\widetilde{T})$ of exact 1-forms, differs by a constant multiple of $d\overline{\phi}^2$. Thus we can characterize $\phi^1 d\overline{\phi}^2$ as the unique antiderivative which vanishes at $\phi^1 = 0$, modulo the group $d\Omega^0(\widetilde{T})$ of exact 1-forms. Geometrically, the lift of $(\pi_T, \nabla_T)$ to $\widetilde{T}$ has a nonflat trivialization whose covariant derivative is $\phi^1 d\overline{\phi}^2$, unique up to a locally constant function. Returning to the maps on $M$, we conclude that the lift of $\overline{\phi}^1 : M \to \mathbb{R}/\mathbb{Z}$ to $\phi^1 : M \to \mathbb{R}$ produces a nonflat trivialization $\phi^1 \cdot \overline{\phi}^2$ of the product $\overline{\phi}^1 \cdot \overline{\phi}^2$.

*Remark* 137. We emphasize a few lessons, which pertain to the product (131) in any degrees as well as to multiplicative generalized differential cohomology theories. First, the nonflat trivialization of one factor implies that the product only depends on the curvature of the other factor. So here the nonflat trivialization $\phi^1$ of $\overline{\phi}^1$ implies that shifting $\overline{\phi}^2$ by a flat element does not alter the product $\overline{\phi}^1 \cdot \overline{\phi}^2$. Furthermore, the product is determined by the image of $\phi^1 d\overline{\phi}^2$ in the vector space $\Omega^1(M)/d\Omega^0(M)$. The second point to emphasize is the last sentence before this remark. We use a "categorified" product $\overline{\phi}^1 \cdot \overline{\phi}^2$ of geometric objects, not just the product $[\overline{\phi}^1] \cdot [\overline{\phi}^2]$ of their isomorphism classes. Therefore, it makes sense to talk about a trivialization of $\overline{\phi}^1 \cdot \overline{\phi}^2$. More sharply, the product $\tau \cdot \lambda$ of a nonflat trivialization $\tau$ of an object $\kappa$ with another object $\lambda$ is a nonflat trivialization of $\kappa \cdot \lambda$.

In another direction, suppose $\overline{\phi}^1 \colon M \to \mathbb{R}/\mathbb{Z}$ is locally constant. Then $d\overline{\phi}^1 = 0$, so the product (134) is the isomorphism class of a flat bundle—the curvature $d\overline{\phi}^1 \wedge d\overline{\phi}^2$ vanishes. Suppose for simplicity of exposition that $M$ is connected, so that $\overline{\phi}^1 = \bar{x}$ is constant. Then the map $\overline{\phi}^1 \times \overline{\phi}^2 \colon M \to T = \mathbb{R}/\mathbb{Z} \times \mathbb{R}/\mathbb{Z}$ factors through the circle $\{\bar{x}\} \times \mathbb{R}/\mathbb{Z} \subset T$. The restriction of $(\pi_T, \nabla_T)$ to this circle is a flat bundle with holonomy $\bar{x} \in \mathbb{R}/\mathbb{Z}$ by the characterization (133) of $(\pi_T, \nabla_T)$ together with Stokes' theorem applied to $[0, x] \times \mathbb{R}/\mathbb{Z}$, where $x \in [0, 1)$ is a lift of $\bar{x} \in \mathbb{R}/\mathbb{Z}$. So the class of this restriction in $H^1\big(\{\bar{x}\} \times \mathbb{R}/\mathbb{Z}; \mathbb{R}/\mathbb{Z}\big)$, under the isomorphism (115), is $\bar{x}$ times the canonical element of $H^1(\mathbb{R}/\mathbb{Z}; \mathbb{R}/\mathbb{Z})$. Hence in this case the product (134) in differential cohomology reduces to the cup product in cohomology:

$$\begin{aligned}
\smile \colon H^0(M; \mathbb{R}/\mathbb{Z}) \times H^1(M; \mathbb{Z}) &\longrightarrow H^1(M; \mathbb{R}/\mathbb{Z}) \\
[\overline{\phi}^1] \quad , \quad c([\overline{\phi}^2]) &\longmapsto [\overline{\phi}^1] \smile c([\overline{\phi}^2]),
\end{aligned} \tag{138}$$

where, as in (114), $c(\cdot)$ denotes the characteristic class, which here is the homotopy class of the map $\overline{\phi}^2$. The cohomological formula (138) also holds when $\overline{\phi}^1$ is locally constant but not constant.

## 5.4 Integration

In general, for $k$ a positive integer, $W$ a closed oriented $k$-dimensional manifold, and $M$ a closed oriented $(k-1)$-dimensional manifold, there are integration maps

$$\int_W \colon \mathcal{A}^k(W) \longrightarrow \mathbb{Z}, \tag{139}$$

$$\int_M \colon \mathcal{A}^k(M) \longrightarrow \mathbb{R}/\mathbb{Z}. \tag{140}$$

The first map (139) is a *primary invariant*. It factors through the characteristic class map

$$c \colon \mathcal{A}^k(W) \longrightarrow H^k(W; \mathbb{Z}) \tag{141}$$

and is computed by evaluation on the fundamental class of $W$ in homology. The second integration (140) is the more interesting *secondary invariant*, and it depends on the geometric data.

**Example 142.** We illustrate with $k = 2$. Suppose $\pi \colon L \to M$ is a hermitian line bundle with compatible covariant derivative $\nabla$, and let $\kappa \in \mathcal{A}^2(M)$ be its class in differential cohomology. If $M$ is a closed oriented surface, then $\int_M \kappa \in \mathbb{Z}$ is the degree of the bundle, the integral of its Chern class, and does not depend on $\nabla$. If $M = S^1$ is an oriented circle, then $\int_{S^1} \kappa \in \mathbb{R}/\mathbb{Z}$ is minus the logarithm of the holonomy of $\nabla$ around the circle. We can also integrate if $M = [0, 1]$

is a compact manifold with boundary. Then we integrate the actual geometric object $(\pi, \nabla)$, not its isomorphism class. The result $\int_{[0,1]}(\pi, \nabla)$ is the parallel transport of $\nabla$ along $[0,1]$, which is an isometry $L_0 \to L_1$ from the fiber of $\pi\colon L \to [0,1]$ over 0 to the fiber over 1.

This last point illustrates that much more general integrations than (139) and (140) are defined. If $M \to S$ is a fiber bundle whose fibers are coherently oriented closed manifolds of dimension $n$, then integration along the fiber in differential cohomology is a map

$$\int_{M/S} : \mathcal{A}^k(M) \longrightarrow \mathcal{A}^{k-n}(S). \tag{143}$$

So, for example, if the fibers have dimension $k - 2$, then the result is an *isomorphism class* of hermitian line bundles with covariant derivative over $S$. A more precise integration starts with a geometric object $\hat{\kappa}$ representing a class $\kappa \in \mathcal{A}^k(M)$; the result is a geometric object $\int_{M/S} \hat{\kappa}$ representing $\int_{M/S} \kappa \in \mathcal{A}^{k-n}(S)$. Furthermore, if $\hat{\tau}$ is a nonflat trivialization of $\hat{\kappa}$, then $\int_{M/S} \hat{\tau}$ is a nonflat trivialization of $\int_{M/S} \hat{\kappa}$. There are also integration maps over fiber bundles of compact manifolds with boundary, and appropriate versions of Stokes' theorem hold.

*Remark* 144. We refer to [46–49] for precise statements, constructions, theorems, and proofs as well as additional references. Presumably more work is needed to develop the full "categorified integration theory" we use here.

**Example 145.** Resuming Example 142, suppose $M \to S$ is a fiber bundle with fiber $S^1$, and $\pi\colon L \to M$ is a hermitian line bundle with covariant derivative $\nabla$. We already asserted that the value of $\bar{f} = \int_{M/S}(\pi, \nabla)\colon S \to \mathbb{R}/\mathbb{Z}$ at $s \in S$ is minus the log holonomy of $\nabla$ around the oriented fiber $M_s$ of $M \to S$ at $s$. Now suppose $\tau\colon M \to L$ is a nonflat trivialization of $(\pi, \nabla)$. Then $f = \int_{M/S} \tau\colon S \to \mathbb{R}$ is a nonflat trivialization of $\bar{f}$, which in this degree simply means that $f \equiv \bar{f} \pmod{\mathbb{Z}}$. It is computed by writing $\frac{\sqrt{-1}}{2\pi}\nabla\tau = \omega_\tau \tau$ for $\omega_\tau \in \Omega^1(M)$; then $f = \int_{M/S} \omega_\tau$.

## 5.5 Generalized Cohomology; Tate Twists

The marriage of integral cohomology and differential forms generalizes to arbitrary cohomology theories, such as $K$-theory. The only caveat is that not every cohomology theory admits products, and the differential theory follows suit. Furthermore, the orientation condition for integration depends on the underlying cohomology theory, but now may require differential geometric data as well. For example, integration in differential $KO$-theory requires a Riemannian metric as part of a "differential orientation"; the topological orientation data is a spin structure, as it is for topological $KO$-theory. If $h$ denotes a generalized cohomology theory, then we use $\check{h}^k(M)$ to notate the differential $h$-group of $M$ in degree $k$. The formal properties of generalized differential cohomology are analogous to those of ordinary differential cohomology.

The secondary invariant (140) is a signature feature of the differential theory not present in the underlying topological theory. If $h$ is $K$-theory and $N$ is an odd-dimensional spin Riemannian manifold, then the secondary invariant is the Atiyah-Patodi-Singer $\eta$-invariant; see [74, 75]. This is important for the anomaly theory of a free spinor field.

A simple example of a cohomology theory beyond ordinary integer cohomology is ordinary cohomology with coefficients in an abelian group. Introduce the abelian groups

$$\mathbb{Z}(m) = (2\pi\sqrt{-1})^m \mathbb{Z}, \qquad m \in \mathbb{Z}^{\geq 0}. \tag{146}$$

For $m$ even $\mathbb{Z}(m) \subset \mathbb{R}$ and for $m$ odd $\mathbb{Z}(m) \subset \sqrt{-1}\mathbb{R}$. Each $\mathbb{Z}(m)$ is an abelian group—a module over $\mathbb{Z}$—and there is a multiplication map

$$\mathbb{Z}(m_1) \times \mathbb{Z}(m_2) \longrightarrow \mathbb{Z}(m_1 + m_2). \tag{147}$$

The *Tate twist* (146) enters naturally for us in two ways. First, exponentiation is a map

$$\exp\colon \mathbb{R}(1)/\mathbb{Z}(1) \longrightarrow \mathbb{C},$$
$$\bar{\zeta} \longmapsto e^{\bar{\zeta}}, \tag{148}$$

where $\mathbb{R}(1) = \sqrt{-1}\mathbb{R}$. This is simply the assertion that $e^{2\pi\sqrt{-1}n} = 1$ for all integers $n$. In the physics models we encounter $\mathbb{R}/2\pi\mathbb{Z}$-valued functions[35] $\theta\colon M \to \mathbb{R}/2\pi\mathbb{Z}$, and so we set $\bar{\zeta} = \sqrt{-1}\theta$ in (148). Then the isomorphism class is

$$[\sqrt{-1}\theta] \in \check{H}^1(M;\mathbb{Z}(1)). \tag{149}$$

Second, the de Rham cohomology class of the curvature of a covariant derivative $\nabla$ on a hermitian line bundle $\pi\colon L \to M$ lies in the image of

$$H^2(M;\mathbb{Z}(1)) \longrightarrow H^2(M;\mathbb{R}(1)). \tag{150}$$

So it is natural to locate

$$[\pi,\nabla] \in \check{H}^2(M;\mathbb{Z}(1)). \tag{151}$$

Then the map (112) gives the curvature on the nose (as an element of $\Omega^2_{\mathrm{closed}}(M;\mathbb{R}(1))$). At the same time we locate the Chern class $c(\pi)$ in $H^2(M;\mathbb{Z}(1))$. The resulting differential cohomology group is denoted $\check{H}^2(M;\mathbb{Z}(1))$.

# 6 Invertible Field Theories

Recall that in section 1.6 we introduced the domain of a field theory. For a noninvertible theory it involves bordism categories of manifolds equipped with background fields. For an invertible theory $\alpha$ we can take it to be a "generalized smooth manifold" $\mathcal{X}$ on which we define generalized differential cohomology objects, and we identify an invertible theory with such an object. For an $n$-dimensional invertible theory the underlying isomorphism class is an element $[\alpha] \in \check{h}^{n+1}(\mathcal{X})$ for some choice of cohomology theory $h$. *Flat* differential cohomology objects correspond to invertible *topological* field theories, and for those we can replace $\mathcal{X}$ by a bordism spectrum in the sense of stable homotopy theory; see [50]. We remark that all theories we encounter in this paper are Wick rotations of *unitary* theories.

We describe a few examples of invertible field theories in section 6.2. First, in section 6.1 we paint an impressionistic picture of a mathematical world in which the domain of a Wick-rotated field theory may be located.

## 6.1 General Picture

Segal [76] initiated a geometric framework for field theory in which the domain is a bordism category of smooth manifolds equipped with additional structure, the background fields. This point of view is highly developed for topological field theories, as for example in [77–79]. The framework applies as well to non-topological theories. If a theory is invertible, then the theory factors through a domain in which all objects and morphisms are invertible [50, 80]. This allows us to move from categories to stable homotopy theory: the domain of an invertible field theory is an *infinite loop space*. For *topological* field theories we can identify this infinite loop space explicitly, and this leads to precise computations with many applications. There are no analogous identification theorems for non-topological theories, but our arguments in this

---

[35]It would be better to take imaginary functions valued in $\mathbb{R}(1)/\mathbb{Z}(1)$, but alas that is not what is done!

paper do not require them. We refer to [81] and the references therein for an exposition of these ideas. Invertible field theories were introduced in [82].

So far we have not built in *smoothness* in the sense that quantities computed in a field theory, such as partition functions and correlation functions, are smooth functions in smooth families of manifolds equipped with background fields. Put differently, we can evaluate field theories on smooth *fiber bundles* of manifolds, not merely on single manifolds, and the result should vary smoothly in the base space. The general mathematical maneuver to incorporate smoothness goes back to Grothendieck: sheafify over the category of smooth manifolds and smooth maps. (See [83] for an exposition.) Loosely speaking, then, the domain of a field theory is a sheaf of higher bordism categories over the site of smooth manifolds. We refer to Stolz-Teichner [84, §2] for further discussion and also remark that this formulation of smoothness also enters the approach to quantum field theory via factorization algebras [85].

**Example 152.** The Wick rotation $F$ of a $d$-dimensional bosonic field theory with no symmetry beyond basic Poincaré invariance is a theory of oriented Riemannian manifolds. So if $X$ is a closed oriented Riemannian manifold of dimension $d$, then $F(X)$ is a complex number. Furthermore, if $X \to S$ is a fiber bundle of such manifolds,[36] then $F(X \to S)$ is a smooth complex-valued function on $S$. The collection of such fiber bundles are (some of the) $d$-morphisms in the higher category attached to $S$ in the domain of the theory $F$.

An *invertible* field theory factors through a domain which is a (pre)sheaf of infinite loop spaces. This brings us to the realm of stable homotopy theory. An infinite loop space is the 0-space of a *spectrum*, and we can use spectra in place of infinite loop spaces. The codomain of an invertible field theory is also a sheaf of spectra. For topological field theories the appropriate choice is the Anderson dual $I\mathbb{Z}$ to the sphere spectrum; see [50, §5.3] and [81, §6] for a justification. However, many theories factor through a simpler spectrum, as they do for the anomaly theories we encounter in this paper. An invertible non-topological field theory takes values in the differential version of $I\mathbb{Z}$. We remark that generalized *differential* cohomology is also a sheaf on the site of smooth manifolds. In summary, then, an invertible field theory is, up to isomorphism, a generalized differential cohomology class on a sheaf of spectra. We give several examples in section 6.2 and apply this framework to anomalies in section 7. In all cases here except for the free fermion, the isomorphism class of the anomaly is an ordinary differential cohomology class.

The inchoate ideas expressed here cry out for a detailed mathematical treatment.

**Example 153.** Denote the sheaf of spectra obtained by group completion of the domain in Example 152 as $\mathcal{M}\mathrm{SO}_{\mathrm{Riem}}$. The script '$\mathcal{M}$' reminds us that this is a sheaf of spectra over smooth manifolds, not a single spectrum. Without the Riemannian metrics the value of the sheaf on a point is the usual Thom bordism spectrum $M\mathrm{SO}$. This explains the notation.

*Remark* 154. If the bordisms are equipped with a function to a fixed smooth manifold $M$, then we denote the sheaf of spectra obtained by group completion as $\mathcal{M}\mathrm{SO}_{\mathrm{Riem}} \times M$. (We ignore basepoints and use ordinary Cartesian product for readability.)

*Remark* 155. In section 1.6 we blurred the distinction between sheaves of higher categories, the domain of noninvertible field theories, and sheaves of spectra, the domain of invertible field theories. In the sequel we only apply the notation to invertible theories.

*Remark* 156. We can define flat and nonflat trivializations of an invertible field theory, just as we did in section 5.2 when working over a single manifold rather than in this sheaf-theoretic context.

---

[36]The fibers are closed $d$-manifolds, the relative tangent bundle is endowed with an orientation and metric, and there is a horizontal distribution.

## 6.2 Three Examples

Here are three examples of invertible field theories to illustrate the formalism.

**Example 157** (holonomy). Fix a smooth manifold $M$, a hermitian line bundle $\pi\colon L \to M$, and a compatible covariant derivative $\nabla$. There is an invertible 1-dimensional field theory $\alpha_1$ of 0- and 1-manifolds equipped with two background fields:

$$
\begin{array}{ll}
\text{(i)} & \text{an orientation}\\
\text{(ii)} & \text{a smooth map to } M.
\end{array}
\tag{158}
$$

The value of $\alpha_1$ on an oriented circle $S^1$ with $\phi\colon S^1 \to M$ is the holonomy of $\nabla$ around the loop $\phi$. It can be computed as

$$
\exp\left(-\int_{S^1}\phi^*(\pi,\nabla)\right),
\tag{159}
$$

where $(\pi,\nabla)$ represents an element of $\check{H}^2(M;\mathbb{Z}(1))$, as in (151), and the integral only depends on the isomorphism class $[\pi,\nabla] \in \check{H}^2(M)$; see Example 142. On the other hand, the value of $\alpha_1$ on a positively oriented point $\mathrm{pt}_+$ with $\phi\colon \mathrm{pt}_+ \to M$ is the hermitian line $L_{\phi(\mathrm{pt}_+)}$, the fiber of $\pi\colon L \to M$ at $\phi(\mathrm{pt}_+)$. It depends on the actual bundle, not just its isomorphism class.

   The classifying object for the data—the domain of $\alpha_1$—is

$$
\mathfrak{X}_1 = \mathcal{M}\mathrm{SO}\times M,
\tag{160}
$$

and $(\pi,\nabla)$ on $M$ pulls back to $(\pi_{\mathfrak{X}_1},\nabla_{\mathfrak{X}_1})$ on $\mathfrak{X}_1$. The sheaf $\mathcal{M}\mathrm{SO}$ is analogous to $\mathcal{M}\mathrm{SO}_{\mathrm{Riem}}$ in Example 153, but there are no Riemannian metrics. It carries a Thom class

$$
U \in \check{H}^0(\mathcal{M}\mathrm{SO};\mathbb{Z}).
\tag{161}
$$

The isomorphism class of $\alpha_1$ is

$$
[\alpha_1] = U \cdot [\pi_{\mathfrak{X}_1},\nabla_{\mathfrak{X}_1}] \in \check{H}^2(\mathfrak{X}_1;\mathbb{Z}(1)).
\tag{162}
$$

Recall from section 6.1 that $\mathfrak{X}_1$ is a sheaf of spectra. The $(-k)$-space evaluated on a smooth manifold $S$ is a collection of fiber bundles of $k$-dimensional manifolds over $S$. The values of (182) and (183) are the integrals over the fibers of the indicated expressions; the results have degree $2 - k$ and lie in the cohomology of $S$. The deformation class of $\alpha_1$ is

$$
U \cdot [\pi_{\mathfrak{X}_1}] \in H^2(\mathfrak{X}_1;\mathbb{Z}(1)).
\tag{163}
$$

*Remark* 164. The theory is labeled by its isomorphism class $[\alpha_1]$, although to construct it we need to choose a particular geometric representative. In higher dimensions there are often no readily accessible geometric models, so we simply tell the isomorphism class in generalized differential cohomology. But the theory requires a choice of representative, not just the isomorphism class.

**Example 165** ($\theta$-term in 2-dimensional abelian gauge theory). This is a 2-dimensional invertible theory $\alpha_2$ on manifolds with three background fields:

$$
\begin{array}{ll}
\text{(i)} & \text{an orientation}\\
\text{(ii)} & \text{a principal } \mathbb{T}\text{-bundle}^{37} \text{ with connection}\\
\text{(iii)} & \text{a smooth map to } \mathbb{R}/2\pi\mathbb{Z}.
\end{array}
\tag{166}
$$

Let $X$ be a closed oriented 2-manifold, $P \to X$ a circle bundle with connection $A$, and $\theta \colon X \to \mathbb{R}/2\pi\mathbb{Z}$ a smooth function. The partition function of $\alpha_2$ on this data is often written

$$\text{``}\exp\left(-\frac{1}{2\pi}\int_X \theta F_A\right)\text{''},\tag{167}$$

where $F_A \in \sqrt{-1}\,\Omega^2(X)$ is the curvature of $A$; see (77). The expression (167) is well-defined if $\theta$ is (locally) constant, in which case the result only depends on the de Rham cohomology class of $F_A$ in $H^2(X;\mathbb{R}(1))$, a multiple of the first Chern class of $P$. For nonconstant $\theta$ use multiplication and integration in differential cohomology (sections 5.3–5.4) to make sense of (167). Tate twists, as in (149) and (151), help keep track of factors of $2\pi$ and $\sqrt{-1}$. Namely, $-\theta/2\pi$ represents a class in $\check{H}^1(X;\mathbb{Z})$ and $(P,\Theta)$ represents a class in $\check{H}^2(X;\mathbb{Z}(1))$; the partition function (167) is the exponential of the integral of their product.

The domain of $\alpha_2$ is

$$\mathcal{X}_2 = \mathcal{M}\mathrm{SO} \times B_\nabla \mathbb{T} \times \mathbb{R}/2\pi\mathbb{Z},\tag{168}$$

where $B_\nabla \mathbb{T}$ is the classifying object for $\mathbb{T}$-connections; see [83]. There are canonical universal classes in $\check{H}^2(B_\nabla\mathbb{T};\mathbb{Z}(1))$ and $\check{H}^1(\mathbb{R}/2\pi\mathbb{Z};\mathbb{Z})$. Their pullbacks to $\mathcal{X}_2$ are classes $[\Theta_{\mathcal{X}_2}] \in \check{H}^2(\mathcal{X}_2;\mathbb{Z}(1))$ and $[-\theta_{\mathcal{X}_2}/2\pi] \in \check{H}^1(\mathcal{X}_2;\mathbb{Z})$ which represent the canonical $\mathbb{T}$-connection on $\mathcal{X}_2$ and the canonical map $\mathcal{X}_2 \to \mathbb{R}/2\pi\mathbb{Z}$, respectively. The isomorphism class of $\alpha_2$ is the product

$$[\alpha_2] = U \cdot [\Theta_{\mathcal{X}_2}] \cdot \left[\frac{-\theta_{\mathcal{X}_2}}{2\pi}\right] \in \check{H}^3(\mathcal{X}_2;\mathbb{Z}(1))\tag{169}$$

in differential cohomology, where $U$ is the Thom class (161). The deformation class is the underlying cohomology class in $H^3(\mathcal{X}_2;\mathbb{Z}(1))$.

*Remark* 170. Pull back this theory to

$$\widetilde{\mathcal{X}}_2 = \mathcal{M}\mathrm{SO} \times B_\nabla\mathbb{T} \times \mathbb{R},\tag{171}$$

via the quotient map $\phi \colon \widetilde{\mathcal{X}}_2 \to \mathcal{X}_2$, so to a theory in which $\theta$ is lifted to an $\mathbb{R}$-valued function $\tilde{\theta}$. Then, analogously to the discussion preceding Remark 137, the lift $\tilde{\theta}_{\widetilde{\mathcal{X}}_2}$ trivializes the pullback $\phi^*\theta_{\mathcal{X}_2}$, and so $\phi^*\alpha_2$ has a canonical trivialization $U \cdot \phi^*\Theta_{\mathcal{X}_2} \cdot (-\theta_{\widetilde{\mathcal{X}}_2}/2\pi)$ on $\widetilde{\mathcal{X}}_2$.

**Example 172** (classical Chern-Simons invariant). The (gravitational) Chern-Simons invariant of a closed oriented Riemannian 3-manifold $X$ is the partition function of an invertible 3-dimensional field theory $\alpha_3$. (Chern and Simons [56] use $\alpha_3(X)$ to derive an obstruction to conformally immersing $X$ into Euclidean 4-space $\mathbb{E}^4$.) It is the secondary invariant of the first Pontrjagin class $p_1$. The background fields are:

$$\begin{array}{l}\text{(i)\ \ an orientation}\\ \text{(ii)\ a Riemannian metric.}\end{array}\tag{173}$$

Hence the domain of $\alpha_3$ is

$$\mathcal{X}_3 = \mathcal{M}\mathrm{SO}_{\mathrm{Riem}},\tag{174}$$

which appears in Example 153. The first Pontrjagin class $p_1$ is a characteristic class of principal $O_n$-bundles for any positive integer $n$. It has a refinement to a *differential* characteristic class $\check{p}_1$ of principal $O_n$-bundles *with connection*. On the level of isomorphism classes this appears in [71], and it reappears in various forms in subsequent works. The integral of $\check{p}_1$ over compact oriented manifolds defines an invertible field theory: classical Chern-Simons theory [86]. Since a Riemannian manifold has a canonical Levi-Civita connection, we can use it to define

$$[\alpha_3] = 2\pi\sqrt{-1}\, U \cdot \check{p}_1 \in \check{H}^4(\mathcal{M}\mathrm{SO}_{\mathrm{Riem}};\mathbb{Z}(1)).\tag{175}$$

---

[37]$\mathbb{T} = \{\lambda \in \mathbb{C} : |\lambda| = 1\}$ is the circle group, also known as $U(1)$.

*Remark* 176. The signature $\text{Sign}(W) \in \mathbb{Z}$ of a closed oriented 4-manifold $W$ satisfies

$$\text{Sign}(W) = \frac{1}{3}\langle p_1(W), [W] \rangle, \tag{177}$$

i.e., the first Pontrjagin class of the *tangent bundle* is divisible by 3. This divisibility is special for the (intrinsic) tangent bundle; it does not hold for arbitrary extrinsic bundles. On a *spin* 4-manifold there is more: $p_1$ is divisible by 48. Atiyah-Patodi-Singer define secondary invariants, called *$\eta$-invariants*, which take advantage of this divisibility. See [87, §4] for the relationship between $\eta$-invariants and Chern-Simons invariants.

# 7 Anomalies and Differential Cohomology

We apply the ideas of section 5 and section 6 to several of the systems discussed earlier in the paper. Our goal here is limited to the derivation and expression of the anomaly theory. In particular, we do not repeat the detailed arguments about dynamical consequences, deformations, and interfaces. The four examples explained here should be sufficient for the reader to work out the others which appear in earlier sections. In the two examples in sections 7.1–7.2 we derive the anomaly directly from the lagrangian. The last example—the massive free spinor field—has an anomaly after performing the fermionic path integral. Here, as usual with spinor fields, geometric index theory determines the precise form of the anomaly. We treat the 4-dimensional theory in section 7.3 by directly applying a theorem of Kahle [51]. In section 7.4 we conjecture a general formula for the isomorphism class of the anomaly of a free spinor field in any dimension.

## 7.1 Particle on a Circle

Consider first the system in section 2. It is a 1-dimensional theory with *fluctuating* scalar field a function $q$ with values in $\mathbb{R}/2\pi\mathbb{Z}$. The theory has a *background* scalar field $\theta$ which also has values in $\mathbb{R}/2\pi\mathbb{Z}$. There is a global $\mathbb{R}/2\pi\mathbb{Z}$-symmetry which acts on $q$ by translation. Finally,[38] there is a time-reversal symmetry which sends $\theta$ to $-\theta$. So if $X$ is a 1-manifold (no orientation), then the background fields on $X$ are:

(i) a Riemannian metric on $X$
(ii) a principal $\mathbb{R}/2\pi\mathbb{Z}$-bundle $P \to X$ with connection[39] $A \in \Omega^1(P)$, and (178)
(iii) a function $\theta \colon X_{w_1} \to \mathbb{R}/2\pi\mathbb{Z}$ such that $\theta \circ \sigma = -\theta$,

where $X_{w_1} \to X$ is the orientation double cover and $\sigma \colon X_{w_1} \to X_{w_1}$ is the non-identity deck transformation. In other words, $\theta$ is a section of the fiber bundle over $X$ with fiber $\mathbb{R}/2\pi\mathbb{Z}$ associated to the orientation double cover $X_{w_1} \to X$ via the action $\theta \mapsto -\theta$ of the cyclic group of order two on $\mathbb{R}/2\pi\mathbb{Z}$. Because of the background field (ii), due to the $\mathbb{R}/2\pi\mathbb{Z}$-symmetry, the fluctuating field $q$ is a section of $P \to X$. The kinetic term in the lagrangian is the "minimal coupling" $\frac{1}{2}|q^*A|^2$, which for the trivial $\mathbb{R}/2\pi\mathbb{Z}$-bundle specializes to the usual $\frac{1}{2}|dq|^2$. Assume $X$ is compact. The anomalous term in the action is the $\theta$-term, written informally in (35) and rendered here as

$$\exp\left( \frac{1}{2\pi\sqrt{-1}} \int_X \theta\, q^*A \right), \tag{179}$$

where we have replaced '$\dot{q} - A$' by the 1-form $q^*A \in \Omega^1(X)$. Let us give meaning to (179) using differential cohomology. The function $\theta$ represents a class in $\breve{H}^1(X; 2\pi\mathbb{Z}_{w_1})$, where

---

[38]There is also a charge conjugation symmetry which flips the signs of $\theta$, $A$, and $q$. We omit it in this account; it can be included by proceeding as in section 7.2.

[39]The Lie algebra of $\mathbb{R}/2\pi\mathbb{Z}$ is $\mathbb{R}$, so $A$ is a *real* 1-form.

$2\pi\mathbb{Z}_{w_1} \to X$ is a local system associated to the orientation double cover. View $q^*A$ as a connection on the topologically trivial $\mathbb{R}/2\pi\mathbb{Z}$-bundle over $X$; in other words, the fluctuating field $q$ is a nonflat trivialization (section 5.2) of $P \to X$ with its connection $A$. The pair $(P,A)$ represents a class in $\check{H}^2(X; 2\pi\mathbb{Z})$. Therefore, the product $\theta \cdot q^*A$ is a nonflat trivialization of the product $\theta \cdot (P,A)$, the latter representing a class in $\check{H}^3(X; \mathbb{Z}(2)_{w_1})$. The anomaly on $X$, then, is the integral[40] of a multiple of this product over $X$.

To write the anomaly as an invertible 2-dimensional field theory $\alpha$, define the domain $\mathcal{X}$ as the total space of the fibering

$$\mathbb{R}/2\pi\mathbb{Z} \longrightarrow \mathcal{X} \longrightarrow \mathcal{M}O_{\text{Riem}} \times B_\nabla(\mathbb{R}/2\pi\mathbb{Z}). \tag{180}$$

Here $B_\nabla(\mathbb{R}/2\pi\mathbb{Z})$ is the classifying object for $\mathbb{R}/2\pi\mathbb{Z}$-connections. For unoriented manifolds the Thom class

$$U \in \check{H}^0(\mathcal{M}O_{\text{Riem}}; \mathbb{Z}_{w_1}) \tag{181}$$

lies in twisted cohomology. Then the product $U \cdot [\theta_{\mathcal{X}}]$ is untwisted and

$$[\alpha] = \frac{1}{2\pi\sqrt{-1}} U \cdot [\theta_{\mathcal{X}}] \cdot [P_{\mathcal{X}}, A_{\mathcal{X}}] \in \check{H}^3(\mathcal{X}; \mathbb{Z}(1)) \tag{182}$$

is the isomorphism class of the anomaly theory, where $A_{\mathcal{X}}$ is the universal connection on the universal bundle $P_{\mathcal{X}} \to \mathcal{X}$. (Compare to section 2.3.) The expression (182) is a nonflat differential cohomology class; its curvature is

$$\frac{1}{2\pi\sqrt{-1}} d\theta_{\mathcal{X}} \wedge dA_{\mathcal{X}} \in \Omega^3(\mathcal{X}; \mathbb{R}(1)). \tag{183}$$

This appears in section 2 as (41).

Knowledge of the anomaly (182) on the domain (180) also gives the value on restricted domains. For example, $\theta_{\mathcal{X}} = \pi$ is a section

$$s: \mathcal{M}O_{\text{Riem}} \times B_\nabla(\mathbb{R}/2\pi\mathbb{Z}) \to \mathcal{X} \tag{184}$$

of the second map in (180), and the pullback anomaly is flat but nontrivial. The constant $\theta_{\mathcal{X}} = \pi$ represents a flat element of $\check{H}^1(\mathcal{M}O_{\text{Riem}} \times B_\nabla(\mathbb{R}/2\pi\mathbb{Z}); 2\pi\mathbb{R}_{w_1}/2\pi\mathbb{Z}_{w_1})$, so an element

$$\pi \in H^0(\mathcal{M}O_{\text{Riem}} \times B_\nabla(\mathbb{R}/2\pi\mathbb{Z}); 2\pi\mathbb{R}_{w_1}/2\pi\mathbb{Z}_{w_1}). \tag{185}$$

As in (138), the differential product (182) is the cup product in ordinary cohomology of this class with the underlying characteristic class $s^*[P_{\mathcal{X}}]$ of $s^*[P_{\mathcal{X}}, A_{\mathcal{X}}]$, where

$$[s^*P_{\mathcal{X}}] \in H^2(MO \wedge B(\mathbb{R}/2\pi\mathbb{Z})_+; \mathbb{Z}(1)) \tag{186}$$

is the first Chern class. These cohomology classes do not depend on metrics and connection, so descend to the familiar spectrum $MO \wedge B(\mathbb{R}/2\pi\mathbb{Z})_+$, which we have rendered with basepoints as usual. Finally, since (185) has order 2, we can express the result as a product in mod 2 cohomology. (The cyclic group of order 2 appears as $\frac{1}{2}\mathbb{Z}(1)/\mathbb{Z}(1)$ in the following.) Since (185) is the nonzero element in the zeroth mod 2 cohomology, the product of (185) and (186) is

$$[s^*\alpha] = \frac{1}{2\pi\sqrt{-1}} \overline{U} \smile [s^*P_{\mathcal{X}}] \in H^2(MO \wedge B(\mathbb{R}/2\pi\mathbb{Z})_+; \tfrac{1}{2}\mathbb{Z}(1)/\mathbb{Z}(1)), \tag{187}$$

where $\overline{U}$ is the mod 2 Thom class. Because the anomaly $s^*\alpha$ is flat, it does not depend on the Riemannian metric or connection: it is an invertible *topological* field theory. Its *deformation*

---

[40]The integral is a hermitian line, and is best thought of over the fibers of a fiber bundle of this data.

class is the twisted Bockstein of (187). We compute it geometrically as follows. Model the deformation class of $\theta_{\mathcal{X}}$, which is a map to $\mathbb{R}/2\pi\mathbb{Z}_{w_1}$, by the principal $2\pi\mathbb{Z}_{w_1}$-bundle of local lifts of $\theta_{\mathcal{X}}$ to $\mathbb{R}_{w_1}$. The universal model for such lifts is the principal $2\pi\mathbb{Z}$-bundle $\mathbb{R} \to \mathbb{R}/2\pi\mathbb{Z}$ with involution $x \mapsto -x$. Over $\theta = \pi$ this restricts to the $2\pi\mathbb{Z}$-torsor $\pi + 2\pi\mathbb{Z} \subset \mathbb{R}$ with free involution $x \mapsto -x$. Mix with the orientation double cover of any smooth manifold $X$ to construct a geometric representative of a cohomology class $\hat{w}_1(X) \in \check{H}^1(X; 2\pi\mathbb{Z}_{w_1})$ which is a twisted integral lift of the first Stiefel-Whitney class. Universally, then, we deduce from (182) that the deformation class of $s^*\alpha$ is:

$$\frac{1}{2\pi\sqrt{-1}} \, U \smile \hat{w}_1 \smile [s^*P_{\mathcal{X}}] \in H^3(MO \wedge B(\mathbb{R}/2\pi\mathbb{Z})_+; \mathbb{Z}(1)). \tag{188}$$

## 7.2 Two-Dimensional $U(1)$ Gauge Theory

This example involves similar considerations to section 7.1, but with some new twists. We discussed a restricted version in Example 165. In this section we incorporate both a charge conjugation symmetry and a time-reversal symmetry.

The theory is described in section 4.1; the anomaly is due to the second term in (77). Since the theory has both time-reversal symmetry and charge conjugation symmetry, it can be formulated on unoriented manifolds equipped with a double cover; the latter is a "gauge field" for the charge conjugation symmetry. The background fields on a 2-manifold $X$ are:

(i) a Riemannian metric;
(ii) a double cover $Q \to X$;
(iii) a function $\theta: X_{w_1} \times Q \to \mathbb{R}/2\pi\mathbb{Z}$ which changes sign under the deck transformations of each of the double covers $X_{w_1} \to X$ (orientation double cover) and $Q \to X$;
(iv) a twisted $U(1)$-gerbe with connection $B$ over $X$, where the twisting is by $Q \to X$.

(189)

The isomorphism classes of $\theta$ and $B$ are located in the twisted cohomology groups

$$\begin{aligned}
[\theta] &\in \check{H}^1(X; \mathbb{Z}(1)_{w_1+Q}) \\
[B] &\in \check{H}^3(X; \mathbb{Z}(1)_Q).
\end{aligned} \tag{190}$$

The differential cohomology product $\theta \cdot B$ is twisted by the orientation double cover and so can be integrated over $X$. (Better: integrate over the fibers of a fiber bundle of such data.) The fluctuating field $a$ is a nonflat trivialization of $B$, so the product $\theta \cdot a$ integrates to a nonflat trivialization of the integral of $\theta \cdot B$. This is the meaning of the second term in (77).

The 3-dimensional anomaly theory $\alpha$ has domain the total space $\mathcal{X}$ of a fibering

$$\mathbb{R}/2\pi\mathbb{Z} \times B_\nabla^2(U(1)) \longrightarrow \mathcal{X} \longrightarrow \mathcal{M}O \times B(\mathbb{Z}/2\mathbb{Z}), \tag{191}$$

where $B_\nabla^2(U(1))$ is the classifying object for $U(1)$-gerbes with connection. Identify $U(1)$ with $\mathbb{R}(1)/\mathbb{Z}(1)$ by exponentiation, and then the universal $U(1)$-gerbe with connection over $B_\nabla^2(U(1))$ represents a class $[\sqrt{-1}B_{\mathcal{X}}] \in \check{H}^3(B_\nabla^2(U(1)); \mathbb{Z}(1))$. With notation parallel to (182), the isomorphism class of $\alpha$ is

$$[\alpha] = \frac{1}{2\pi\sqrt{-1}} \, U \cdot [\theta_{\mathcal{X}}] \cdot [\sqrt{-1}B_{\mathcal{X}}] \in \check{H}^4(\mathcal{X}; \mathbb{Z}(1)), \tag{192}$$

where as in (181) the Thom class lies in $w_1$-twisted cohomology.

As in (184), we can pullback to other domains. For example, to reproduce the anomaly between charge conjugation and the $BU(1)$-symmetry expressed in (93), let $\mathcal{X}'$ be the total space of the fibering

$$B_\nabla^2(U(1)) \longrightarrow \mathcal{X}' \longrightarrow \mathcal{M}SO \times B(\mathbb{Z}/2\mathbb{Z}), \tag{193}$$

where the $B$-field twists as above. Now pull back by the map $\mathcal{X}' \to \mathcal{X}$ which sets $\theta = \pi$ and forgets the orientation. The pullback of $\alpha$ is flat of order 2; the isomorphism class is

$$\frac{1}{2}\overline{U} \smile [B_{\mathcal{X}}] \in H^3\left(\mathcal{X}'; \tfrac{1}{2}\mathbb{Z}(1)/\mathbb{Z}(1)\right) \tag{194}$$

and the deformation class is its integral Bockstein

$$\frac{1}{2\pi\sqrt{-1}} U \smile \widehat{[Q]} \smile [B_{\mathcal{X}}] \in H^4\left(\mathcal{X}'; \mathbb{Z}(1)\right), \tag{195}$$

where $\widehat{[Q]} \in H^1(\mathcal{X}'; 2\pi\mathbb{Z}_Q)$ is the twisted integer lift of the class of $Q$; see also (93).

The computation of the anomaly across an interface, as in the discussion at the end of section 4.1.2, is integration in differential cohomology. We will not comment further.

*Remark* 196. The various symmetries—internal and external—in 2-dimensional $U(1)$ gauge theory can be expressed as a *2-group* $\mathcal{G}$, which sits in a nontrivial fibering

$$BU(1) \times \mathbb{Z}/2\mathbb{Z} \longrightarrow \mathcal{G} \longrightarrow O(2) \times \mathbb{Z}/2\mathbb{Z}. \tag{197}$$

## 7.3 Massive Spinor Fields, Part 1

The interaction between geometric index theory and anomalies of fermionic fields has a long history, almost exclusively focused on the massless case. Here we indicate how to apply Quillen's *superconnections* [88] to compute the precise anomaly in the massive case. In this section we treat the 4-dimensional theory (section 3.3), since it fits cleanly into existing work in geometric index theory [51]. We make some remarks about the 3-dimensional case as well, but further work is needed to complete the story in odd dimensions in these terms. In section 7.4 we conjecture a formula for the anomaly theory in the general case.

There is a unique irreducible real spin representation $\mathbb{S}$ of the Lorentz spin group $\mathrm{Spin}(1,3) \cong \mathrm{SL}(2;\mathbb{C})$. It has real dimension 4; its complexification is $\mathbb{S} \otimes \mathbb{C} \cong \mathbb{S}' \oplus \mathbb{S}''$, where $\mathbb{S}', \mathbb{S}''$ have complex dimension 2 and are complex conjugate. There is an invariant complex skew-symmetric bilinear form on each of $\mathbb{S}', \mathbb{S}''$, so a *complex* line $M(\mathbb{S})$ of invariant real skew-symmetric bilinear forms on $\mathbb{S}$. Fix a nonzero vector $M \in M(\mathbb{S})$, so express an arbitrary element $m \in M(\mathbb{S})$ is $m = (m/M)M$ for $m/M \in \mathbb{C}$. Wick rotate to a closed Riemannian spin 4-dimensional manifold $X$, and let $m: X \to M(\mathbb{S})$ be a smooth function, a variable mass. There are rank 2 complex spin bundles $S^+(X), S^-(X) \to X$. The free spinor field action is a skew-symmetric bilinear form $\omega_X(m)$ on sections of

$$S^+(X) \oplus S^-(X) \longrightarrow X. \tag{198}$$

It is constructed from the Dirac operator and the mass function. The fermionic path integral is

$$\mathrm{pfaff}\,\omega_X(m) \in \mathrm{Pfaff}\,\omega_X(m), \tag{199}$$

an element of the complex Pfaffian line. For a parametrized family of data, encoded in a fiber bundle over a smooth manifold $S$, we obtain a Pfaffian line bundle $\mathrm{Pfaff} \to S$ with hermitian metric and compatible connection. (See [89] and references therein for the massless case.) Its equivalence class in $\breve{H}^2(S; \mathbb{Z}(1))$ is part of the anomaly theory, a 5-dimensional invertible field theory of spin manifolds equipped with a complex-valued function.

We apply geometric index theory to compute the curvature of $\mathrm{Pfaff} \to S$. Consider the $\mathbb{Z}/2\mathbb{Z}$-graded vector bundle which is the tensor product

$$\left[S^+(X) \oplus S^-(X)\right] \otimes \underline{\mathbb{C}^{1|1}} \longrightarrow X, \tag{200}$$

where $\underline{\mathbb{C}}^{1|1} \to X$ is the trivial $\mathbb{Z}/2\mathbb{Z}$-graded complex vector bundle with fiber $\mathbb{C}^{1|1} = \mathbb{C} \oplus \mathbb{C}$. The spinor bundle carries the Levi-Civita covariant derivative. Endow $\underline{\mathbb{C}}^{1|1} \to X$ with the superconnection

$$\boldsymbol{\nabla}(m) = \begin{pmatrix} d & \overline{m/M} \\ m/M & d \end{pmatrix}. \tag{201}$$

Then the *square* of the Pfaffian (199) is the determinant of the Dirac operator[41] coupled to $\boldsymbol{\nabla}(m)$ in the sense of [90, §3.3]. Kahle [51, §6.3] computes $\sqrt{-1}/2\pi$ times the curvature of the determinant bundle for a family of these Dirac operators on a fiber bundle over $S$ as the integral over the fibers of the 6-form component of the differential form

$$\frac{-1}{4\pi^2} \hat{A} \wedge \mathrm{ch}\big(\boldsymbol{\nabla}(m)\big), \tag{202}$$

where $\hat{A}$ is the $\hat{A}$-form[42] of the relative tangent bundle and $\mathrm{ch}\big(\boldsymbol{\nabla}(m)\big)$ is the Chern character form of the superconnection. We must divide this by 2 to account for the determinant line bundle being the square of the Pfaffian line bundle. Then a straightforward computation produces the 6-form

$$\frac{1}{192\pi^2} e^{-|m/M|^2} p_1 \wedge d(m/M) \wedge d(\overline{m/M}), \tag{203}$$

where $p_1$ is the Chern-Weil form of the first Pontrjagin class of the relative tangent bundle. This gives a specific value to $\gamma(m)$ in (3.25).

*Remark* 204. The choice of $M \in M(\mathbb{S})$ sets a scale for the mass. The form (203) peaks around $m = 0$ as $M \to 0$ and flattens out as $M \to \infty$.

*Remark* 205. We explain briefly why odd-dimensional massive spinor fields are not covered by the theorems in [51] in a similar way. For definiteness consider the 3-dimensional case, as in section 3.2. The minimal real representation of the Lorentz spin group $\mathrm{Spin}(1,2) \cong \mathrm{SL}(2;\mathbb{R})$ has dimension 2 and admits a real line of invariant skew-symmetric bilinear forms. Fix a basis $M$ for that line. Under Wick rotation the spin representation complexifies but the mass remains real. Let $X$ be a closed Riemannian spin 3-dimensional manifold and $m/M : X \to \mathbb{R}$ a real-valued function. In this case the appropriate Dirac operator acts on sections of a $\mathbb{Z}/2\mathbb{Z}$-graded bundle of Clifford modules equipped with a *Clifford superconnection* in the sense of [90, Definition 3.39(2)]. Let $S(X) \to X$ be the rank 2 complex spin bundle. Then $S(X) \oplus S(X) \to X$ is a $\mathbb{Z}/2\mathbb{Z}$-graded Clifford module for the bundle of Clifford algebras. Use the mass function and Levi-Civita covariant derivative $\nabla$ to define the superconnection

$$\begin{pmatrix} \nabla & m/M \\ m/M & \nabla \end{pmatrix}. \tag{206}$$

The associated Dirac operator is

$$\begin{pmatrix} \psi^0 \\ \psi^1 \end{pmatrix} \longmapsto \begin{pmatrix} D\psi^0 + (m/M)\psi^1 \\ D\psi^1 + (m/M)\psi^0 \end{pmatrix} \tag{207}$$

on sections of $S(X) \oplus S(X) \to X$. The important point is that the superconnection (206) is *not* induced from a *twisting superconnection*, as in the 4-dimensional case. Berline-Getzler-Vergne [90, Proposition 3.40(2)] prove that in even dimensions the Dirac operator can always be expressed in terms of a twisting superconnection, but as we see here that is not true in odd dimensions. We give an alternative approach in section 7.4.

---

[41]Kahle uses the self-adjoint Dirac operator, as do we in this section. Accordingly, the appropriate Chern character is $\mathrm{Tr}_s\, e^{-\overline{\mathbb{W}}^2}$. In section 7.4 we use different conventions.

[42]Use the Levi-Civita covariant derivative and Chern-Weil theory.

### 7.4 Massive Spinor Fields, Part 2

Here we sketch a conjectural formula for the anomaly theory of a massive spinor field in any dimension. We hope to develop the necessary mathematics elsewhere.

The algebraic theory of massless and massive spinor fields is discussed in [50, §9.2], in part following [91, §6]; further details appear in [92, Appendix]. Let $\mathbb{S}$ be a real spin representation of the Lorentz group $\mathrm{Spin}(1, d-1)$. By definition $\mathbb{S}$ is an ungraded module for the even Clifford algebra[43] $\mathrm{Cliff}(d-1, 1)^0$. With a contractible choice it extends to a $\mathbb{Z}/2\mathbb{Z}$-graded $\mathrm{Cliff}(d-1, 1)$-module structure on $\mathbb{S} \oplus \mathbb{S}^*$. A *mass pairing* is a $\mathrm{Spin}(1, d-1)$-invariant skew-symmetric bilinear form[44]

$$m \colon \mathbb{S} \times \mathbb{S} \longrightarrow \mathbb{R}. \tag{208}$$

Let $M(\mathbb{S})$ denote the real vector space of mass pairings. (It can be the zero vector space.)

*Remark* 209. If $m$ is nondegenerate, then it determines a $\mathrm{Cliff}(d-1, 2)$-module structure on $\mathbb{S} \oplus \mathbb{S}^*$ which extends the given $\mathrm{Cliff}(d-1, 1)$-module structure; see [50, Lemma 9.55].

**Example 210.** For $d = 3$ the spin group is $\mathrm{Spin}(1, 2) \cong \mathrm{SL}(2; \mathbb{R})$, and the unique irreducible real spin representation, which is used in §3.2, is $\mathbb{S} = \mathbb{R}^2$. Identify $\mathbb{S}^* \cong \mathbb{R}^2$ by

$$\begin{pmatrix} 1 & 0 \end{pmatrix} \longleftrightarrow \begin{pmatrix} 0 \\ 1 \end{pmatrix}, \qquad \begin{pmatrix} 0 & 1 \end{pmatrix} \longleftrightarrow \begin{pmatrix} -1 \\ 0 \end{pmatrix}. \tag{211}$$

Let the Clifford generators act on $\mathbb{S} \oplus \mathbb{S}^*$ by

$$e_1 = \left(\begin{array}{c|c} & \begin{matrix} 1 & \\ & -1 \end{matrix} \\ \hline \begin{matrix} 1 & \\ & -1 \end{matrix} & \end{array}\right), \qquad e_2 = \left(\begin{array}{c|c} & \begin{matrix} & 1 \\ 1 & \end{matrix} \\ \hline \begin{matrix} 1 & \\ & 1 \end{matrix} & \end{array}\right), \qquad f = \left(\begin{array}{c|c} & \begin{matrix} & -1 \\ 1 & \end{matrix} \\ \hline \begin{matrix} -1 & \\ 1 & \end{matrix} & \end{array}\right). \tag{212}$$

A mass pairing $m \in M(\mathbb{S}) \cong \mathbb{R}$, written as a linear map $\mathbb{S} \to \mathbb{S}^*$ using the bases (211), is

$$m = \begin{pmatrix} m/M & 0 \\ 0 & m/M \end{pmatrix}, \qquad m/M \in \mathbb{R} \tag{213}$$

relative to a nonzero vector $M \in M(\mathbb{S})$.

Now we sketch a formula for the isomorphism class of the $(d + 1)$-dimensional anomaly theory $\alpha$. The domain is

$$\mathfrak{X} = \mathcal{M}\mathrm{Spin}_{\mathrm{Riem}} \times M(\mathbb{S}), \tag{214}$$

where $\mathcal{M}\mathrm{Spin}_{\mathrm{Riem}}$ is the spin analog of Example 153. Let

$$\underline{\mathbb{S} \oplus \mathbb{S}^*} \longrightarrow M(\mathbb{S}) \tag{215}$$

be the trivial bundle of $\mathrm{Cliff}(d-1, 1)$-modules. Define the superconnection

$$\boldsymbol{\nabla} = d + \hat{m}, \tag{216}$$

where $\hat{m}$ is an odd endomorphism of (215) constructed[45] from $m \in M(\mathbb{S})$. The bundle (215) with superconnection $\boldsymbol{\nabla}$, after pulling back to $\mathfrak{X}$, is a geometric model for an element

---

[43]$\mathrm{Cliff}(p, q)$ has $p$ generators with square $+1$ and $q$ generators with square $-1$.

[44]In this paper we do not require $m$ to be nondegenerate.

[45]Use the generator $f \in \mathrm{Cliff}(d-1, 1)$ with $f^2 = -1$ to define the positive definite inner product

$$(s_1, s_2)_{\mathbb{S}} := \langle f \cdot s_1, s_2 \rangle, \qquad s_1, s_2 \in \mathbb{S}. \tag{217}$$

The mass $m$ defines a linear map $\tilde{\mu} \colon \mathbb{S} \to \mathbb{S}^*$; its adjoint with respect to (217) is a linear map $\tilde{\mu}^* \colon \mathbb{S}^* \to \mathbb{S}$. Then

$$\hat{m} = \left(\begin{array}{c|c} & -\tilde{\mu}^* \\ \hline \tilde{\mu} & \end{array}\right). \tag{218}$$

$[\boldsymbol{\nabla}] \in \widetilde{KO}^{d-2}(\mathcal{X})$. There is a differential orientation [46,75] in differential $K$-theory expressed as the differential Thom class

$$\check{U} \in \widetilde{KO}^{0}(\mathcal{M}\mathrm{Spin}_{\mathrm{Riem}}). \tag{219}$$

Its curvature is the $\hat{A}$-form, and is responsible for the differential form $p_1$ in (226) below. It is the differential version of the Atiyah-Bott-Shapiro map. The isomorphism class of the anomaly theory $\alpha$ is, conjecturally,

$$[\alpha] = \mathrm{Pfaff}\left(\check{U} \cdot [\boldsymbol{\nabla}]\right) \in \widetilde{I\mathbb{Z}(1)}^{d+2}(\mathcal{X}), \tag{220}$$

where $I\mathbb{Z}(1)$ is the Anderson dual to the sphere spectrum and $\mathrm{Pfaff}\colon KO \to \Sigma^4 I\mathbb{Z}(1)$ is the map which gives the Anderson self-duality of $KO$. This generalizes [50, Conjecture 9.63].

*Remark* 221. As it stands, (220) is not sufficient since $M(\mathbb{S})$ is contractible. Rather, we use the geometric models to lift to a relative class. Namely, Remark 209 implies that $(\underline{\mathbb{S} \oplus \mathbb{S}^*}, \boldsymbol{\nabla})$ has a canonical nonflat trivialization over the subset $M(\mathbb{S})^o \subset M(\mathbb{S})$ of nondegenerate mass pairings. (This uses [93] and [94, §4].) The nonflat trivialization lifts the deformation class of the anomaly—the cohomology class underlying (220)—to a relative class in $I\mathbb{Z}(1)^{d+2}(\mathcal{X}, \mathcal{X}^o)$, where $\mathcal{X}^o = \mathcal{M}\mathrm{Spin}_{\mathrm{Riem}} \times M(\mathbb{S})^o$.

*Remark* 222. Here we use *skew-adjoint* Dirac operators, as in [92, Appendix A]. Accordingly, the mass endomorphism $\hat{m}$ is skew-adjoint and the Chern character (225) below has a plus sign in the exponential.

*Remark* 223. The partition function of the anomaly theory $\alpha$ on a closed spin Riemannian $(d+1)$-manifold is the exponentiated $\eta$-invariant of an appropriate Dirac operator.

**Example 224.** Resuming Example 210 we compute the curvature of (220) for $d = 3$ and the minimal choice of $\mathbb{S}$. The Chern character of $\boldsymbol{\nabla}$ is the supertrace of the exponentiated curvature, after acting by the volume form:

$$
\begin{aligned}
\mathrm{Tr}_{\mathrm{s}}\left(e_1 e_2 f\, e^{\boldsymbol{\nabla}^2}\right) &= \mathrm{Tr}_{\mathrm{s}}\, e^{-(m/M)^2}\left(\begin{array}{c|c} d(m/M) & 1 \\ \hline 1 & -d(m/M) \end{array}\right) \\
&= 4 e^{-(m/M)^2} d(m/M).
\end{aligned} \tag{225}
$$

So the 5-form curvature, up to a numerical factor, is

$$e^{-(m/M)^2} d(m/M) \wedge p_1. \tag{226}$$

This tells what $f$ is in (65); the numerical factor $1/\sqrt{\pi}$ normalizes the integral over $\mathbb{R}$.

## Acknowledgements

We thank David Ben-Zvi, Thomas Dumitrescu, Jeffrey Harvey, Mike Hopkins, Po-Shen Hsin, Zohar Komargodski, Gregory Moore, Andy Neitzke, Kantaro Ohmori, Shu-Heng Shao, Ryan Thorngren, and Edward Witten for discussions. The work of H.T.L. is supported by a Croucher Scholarship for Doctoral Study, a Centennial Fellowship from Princeton University and the Institute for Advanced Study. The work of C.C. and N.S. is supported in part by DOE grant de-sc0009988. The work of D.S.F. is supported by the National Science Foundation under Grant Number DMS-1611957. Any opinions, findings, and conclusions or recommendations expressed in this material are those of the authors and do not necessarily reflect the views of the National Science Foundation. C.C. and D.S.F. also thank the Aspen Center for Physics, which is supported by National Science Foundation grant PHY-1607611, for providing a stimulating atmosphere.

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
