# Peer review of "Anomalies in the Space of Coupling Constants and Their Dynamical Applications I"

_SciPost Physics, doi:SciPost Phys. 8, 001 (2020)_

## Round 2 · Referee Report · Anonymous · 2019-9-17

Strengths

1. 't Hooft anomaly matching is generalized, and this can provide more possibilities to constrain non-trivial IR dynamics of strongly-coupled field theories.
2. The paper includes many pedagogical examples.

Weaknesses

Nothing

Report

Anomaly matching provides a useful tool to study strongly coupled field theories thanks to the fact that it is RG invariant, and anomaly matching rules out the trivially gapped vacuum from its possible ground states. Even if the theory does not have any ’t Hooft anomaly, one can sometimes show that there must be a non-trivial IR dynamics when one varies the coupling constants if one can show that different parameter regions are distinguished as SPT phases. This paper is a trial to give a systematic way to treat such cases as a generalization of ’t Hooft anomaly matching.

The paper is well written. It includes sufficiently many pedagogical examples how such a thing is possible, and gives the motivation to formalize the way to prove non-triviality of IR physics without conventional ’t Hooft anomalies. After Sec. 5, mathematical details are explained and it gives a tool to fully formulate the systematic treatment of generalized ’t Hooft anomaly, and this part requires good familiarity with abstract mathematical techniques. But, every idea of this paper is explained before those sections, and I think it is understandable to the broad audience. For example, we can find that many of important formulas in Sec. 7 already appear with physicists’-friendly language in Sec. 4 or before.

I have some questions, which is described below, in order to understand the technical parts of this paper, but, anyway, I strongly recommend the publication of this paper.

Questions

1. I am confused with notation among (1.16), (1.18) and (1.19), in which domain of field theory is discussed.
When I look at (1.19), I understand that the most right of sequence is the space of background metric/gauge fields/spacetime-dependent theta parameter, while the most left is the space of dynamical fields. And, this looks to be important in Remark 1.20 in order to claim that anomaly is ’t Hooft anomaly, which does not depend on dynamical fields.
When I look at (1.16) and (1.18), however, the space of dynamical U(1) gauge fields $B_{\nabla}U(1)$ is put on the right. How should I understand this difference?

2. Does the Scenario Two in the bottom of p.15 correspond to adding spectator fermions as in original argument by ’t Hooft for anomaly matching?

3. In the bottom of p.22, the usefulness of generalizing Stora-Zumino descent procedure is mentioned, and it is claimed that this two-step inflow is generally possible for infinite order anomaly. What about finite order case? Is it always impossible?

  • validity: top
  • significance: top
  • originality: high
  • clarity: high
  • formatting: excellent
  • grammar: perfect

Author:  Ho Tat Lam  on 2019-10-29  [id 634]

(in reply to Report 1 on 2019-09-17)
Category:
answer to question

Question 1 is a good one, and hopefully the following is helpful. First, at each stage in the discussion the fields are all background. Now we may choose to designate a subspace fluctuating and integrate over them, if we can, to obtain a new theory, and that new theory will have a new set of background fields. Now in general the fields form an iterated fibering; the examples given in (1.16) - (1.19) are one-step fiberings. We chose to write them so that the fields in the fibers truly twist over the base. (For a fibering (F --> E --> B) x M we can put M in either the fiber F x M --> E x M --> B or we can put it in the base F --> E x M --> B x M.) The principle illustrated in (1.19) is that if we designate certain fields as fluctuating, say the fields in the fiber of A' --> A --> A'', then we can still (formally) execute the path integral if there is an anomaly which is pulled back from the base A'', and that is what's called an 't Hooft anomaly.

The answer to Question 2 is "no". One example of Scenario Two is to lift a background circle-valued scalar (X) to a real-valued scalar (X'), which in some theories gives a trivialization of an anomaly. Another case is that X' is a subspace of X. For example, in the 2d WZW anomaly there is a global G x G symmetry which is anomalous and there are "anomaly-free" subgroups which we can pull back to.

---

## Round 2 · Referee Report · Anonymous · 2019-10-16

Strengths

1. Notion of 't Hooft anomaly is extended to involve space of coupling constants
2. The generalized anomalies are more robust under deformations than standard 't Hooft anomalies and there are explicit examples where the generalized anomalies give stronger constraints
3. Mathematical perspetives on these anomalies are also provided

Weaknesses

None

Report

't Hooft anomalies have provided nontrivial constraints on dynamics and phases of quantum field theories. In particular any theory with 't Hooft anomalies cannot flow to a trivially gapped theory with a unique vacuum. It is also practically important that 't Hooft anomalies can be computed even in complicated theories which are not solvable by current technology (e.g. QCD). Therefore discovery of a new anomaly is highly appreciated.

This paper generalizes the notion of 't Hooft anomalies to involve space of coupling constants. The basic idea is well explained in terms of the pedagogical example of a particle on circle in section 2. In that example, the generalized anomaly involves theta angle as well as U(1) background gauge field.
The new anomaly can be used to discuss that the system must have a non-unique ground state at least one value of \theta
even if we add a large number of deformations preserving Z_N subgroup of the U(1) symmetry. For deformation preserving charge conjugaration C, the constraint by the new anomaly is
weaker than the one from mixed anomaly between Z_N symmetry and C at \theta=\pi in the sense that the new anomaly does not specialize \theta=\pi. However, the mixed anomaly argument cannot be applied for more general deformations breaking C while the new anomaly is still available. This is the advantage of the new anomaly.

The authors also dicusss other examples such as massive fermions in 1,2 and 4 dimensions, and two dimensional scalar QED. In the massive fermion examples, 't Hooft anomalies are extended to involve space of fermion mass which is real in 2d and 3d, and complex in 4d.

The latter sections discuss mathematical perspectives of the generalized anomalies above in terms of differential cohomology. Although it would be hard to read these sections for ordinary physicists, they should be useful for mathematicians and physicists familiar with mathematical knowledge used there. I do not think that this is a problem since earlier sections already discuss essentially the same physics
in a language familiar with physicists.

In coclusion I think that this paper is well-written and I strongly recommend this paper for publication in SciPost.

Requested changes

A (minor) modification of the following point would be useful for readers:
When I first read the arguments on the descent procedures of the generalized anomalies (footnote 6, around eq.(2.16),(3.9) etc), I was a bit confused on the convention.
It would be better to write explicitly what are \alpha and \omega around eq.(2.16), and a relation between anomaly action and \omega (perhaps 2\pi*i \omega ?).

A typo:
In the last line of page 25, "give" -> "gives"

A possibly typo:
I guess there should be a minus sign in eq.(3.18) to be consistent with eq.(3.15).

---

## Editorial Decision

published